# G-Adaptivity: optimised graph-based mesh relocation for finite element methods

James Rowbottom [* 1]   Georg Maierhofer [* 1]   Teo Deveney [2 3]   Eike Mueller [3]   Alberto Paganini [4]
Katharina Schratz [5]   Pietro Liò [6]   Carola-Bibiane Schönlieb [1]   Chris Budd [3]

## Abstract

We present a novel, and effective, approach to achieve optimal mesh relocation in finite element methods (FEMs). The cost and accuracy of FEMs is critically dependent on the choice of mesh points. Mesh relocation (r-adaptivity) seeks to optimise the mesh geometry to obtain the best solution accuracy at given computational budget. Classical r-adaptivity relies on the solution of a separate nonlinear "meshing" PDE to determine mesh point locations. This incurs significant cost at remeshing, and relies on estimates that relate interpolation- and FEM-error. Recent machine learning approaches have focused on the construction of fast surrogates for such classical methods. Instead, our new approach trains a graph neural network (GNN) to determine mesh point locations by directly minimising the FE solution error from the PDE system Firedrake to achieve higher solution accuracy. Our GNN architecture closely aligns the mesh solution space to that of classical meshing methodologies, thus replacing classical estimates for optimality with a learnable strategy. This allows for rapid and robust training and results in an extremely efficient and effective GNN approach to online r-adaptivity. Our method outperforms both classical, and prior ML, approaches to r-adaptive meshing. In particular, it achieves lower FE solution error, whilst retaining the significant speed-up over classical methods observed in prior ML work.

---

[*]Equal contribution [1]Department of Applied Mathematics and Theoretical Physics, University of Cambridge, UK [2]Department of Computer Science, University of Bath, UK [3]Department of Mathematical Sciences, University of Bath, UK [4]School of Computing and Mathematical Sciences, University of Leicester, UK [5]Laboratoire Jacques-Louis Lions, Sorbonne Université, France [6]Department of Computer Science and Technology, University of Cambridge, UK. Correspondence to: James Rowbottom <jr908@cam.ac.uk>, Georg Maierhofer <gam37@cam.ac.uk>.

*Proceedings of the 42^nd International Conference on Machine Learning*, Vancouver, Canada. PMLR 267, 2025. Copyright 2025 by the author(s).

## 1. Introduction

Finite element methods (FEM) are currently the most widely-used tool for the large scale solution of partial differential equations (PDEs) (Ainsworth & Oden, 1997; Cotter, 2023). Central advantages are robustness, reliable error estimates, and thoroughly developed code bases (such as deal.II (Africa et al., 2024), DUNE (Bastian et al., 2008), Fenics (Logg et al., 2012), and Firedrake (Ham et al., 2023)), which are highly parallelisable and efficient. However, even with such optimised software the simulation of large scale problems (e.g. weather forecasting, structural simulations in engineering systems) is computationally costly. An important ingredient that determines the cost is the number of degrees of freedom (DOFs) required by a FEM to satisfy a chosen error tolerance. Since this cost depends on the number $N_z$ of mesh points $\mathbf{z}^{(i)}$ of the underlying computational mesh, it is desirable to keep $N_z$ moderate. Mesh adaptivity based on mesh refinement and/or relocation to capture important solution features at the right scale can balance computational cost and accuracy. However, classical mesh-adaptive methods can be difficult to implement and require significant computational resources. In contrast, in the present work, we introduce a cheap, stable and highly efficient graph neural network (GNN) architecture to implement learnable mesh relocation ($r$-adaptivity). Our method keeps $N_z$ fixed and adapts the mesh point locations to reduce the overall FE error. Many classical mesh relocation methods have focused on finding and minimising mathematical substitutes of the FE error (usually simplified upper bounds) and solving additional (differential) equations to relocate the mesh points. For example, one can solve the Monge-Ampère (MA) equation (Budd et al., 2009) to find mesh point locations that minimise the interpolation error, which is an upper bound (up to some parameters and constants) of the FEM error arising from Céa's lemma (Huang & Russell, 2011). Recent Machine Learning (ML) approaches rely on similar mathematical simplifications and learn a surrogate for the mesh equations (Song et al., 2022; Zhang et al., 2024) leading to significant speed-up with comparable error reduction. In the present work we take an entirely different approach. We present G-adaptivity, an approach to mesh adaptivity that trains a GNN to generate meshes that *directly minimise the error of the corresponding FEM solution*. We

couple backpropagation through a novel diffusion-based GNN-deformer, with mesh point gradients obtained through an application of Firedrake adjoint (Mitusch et al., 2019; Ham et al., 2023), to minimise the FEM approximation error directly (as opposed to the upper bound considered in (Huang & Russell, 2011)). The result is a model capable of outperforming the current state-of-the-art $r$-adaptive methods, whilst retaining the significant acceleration of ML based approaches (cf. Figure 1).

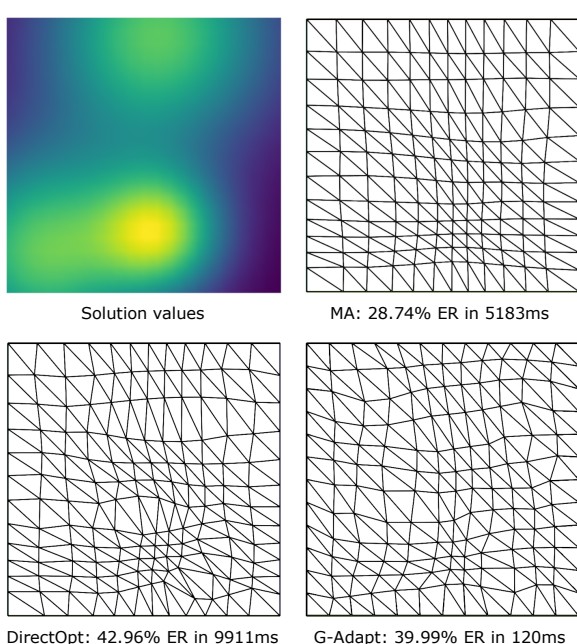

Solution values      MA: 28.74% ER in 5183ms

DirectOpt: 42.96% ER in 9911ms      G-Adapt: 39.99% ER in 120ms

*Figure 1.* Optimised meshes from our new approach (G-Adapt) on the example of Poissons' equation: the error reduction (ER) achieved by classical Monge-Ampère (MA) can be significantly improved with direct optimisation (DirectOpt) of the FEM loss with respect to the mesh points, but at prohibitive additional cost. Our new approach achieves near optimal meshes in a fraction of the inference time.

**Contributions** Our work improves earlier ML based approaches to mesh relocation in the following ways:

- A novel training mechanism capitalising recent advances in FEM systems, which leads to a fast meshing algorithm that reduces the FE error even over state-of-the-art classical meshing methods. This was not possible in any prior surrogate ML approach;

- An improved GNN architecture based on a diffusion deformer, which allows for improved mesh relocation quality and provable non-tangling of generated meshes;

- A novel equidistribution loss regularizer, which enforces mesh regularity in unsupervised GNN training;

- Thorough numerical comparison with classical and recent approaches in terms of accuracy, mesh quality and computational time. Our experiments include both stationary and time-dependent test cases.

## 2. Related work

**The effective approximation of PDE solutions** is one of the central problems in computational mathematics. Over the recent decade, extensive work has been devoted to using ML for the numerical approximation of PDEs. This includes physics informed neural networks (PINNS) (Raissi et al., 2019; Raissi, 2018), Fourier Neural Operators (FNOs) (Li et al., 2020b; 2023), graph neural operators (Li et al., 2020a), DeepONets (Lu et al., 2021), Message Passing Neural PDE Solvers (Brandstetter et al., 2022) and the deep Ritz method (E & Yu, 2018). The majority of such approaches try to directly approximate the PDE, or the associated solution operator, with a machine learning surrogate. Such methods offer certain advantages (for example in high dimensional settings (Han et al., 2018)), but are typically outperformed by traditional numerical methods in accuracy in most settings (Grossmann et al., 2023). Our approach is different. We use ML as a central ingredient of a finite element discretisation to construct an improved computational mesh, which is then coupled to a classical PDE solver. The crucial advantage is that we retain convergence guarantees and robustness of FEMs, something that is often lacking in direct ML-based PDE approximations. At the same time our approach achieves a significant speed up in the calculation of the improved mesh compared to classical approaches.

**Adaptive mesh methods** are a widely used tool for improving the performance of a classical FEM by varying the local density of the mesh points. This is necessary if the PDE solution has small length scales or singularities. Adaptivity allows achieving high accuracy without resorting to uniform mesh refinement. The most popular form is $h$-adaptivity (Ainsworth & Oden, 1997), in which mesh cells are subdivided when an a-posteriori estimate of the solution error is large. Such methods have complex data structures (see e.g. (Burstedde et al., 2011)) and, possibly, poor mesh regularity. Alternatively, the relocation based $r$-adaptive methods considered in this paper move a fixed number of mesh points to achieve a high density of points where a monitor $m(\mathbf{z})$ of the solution error is large. Done correctly this can lead to significant error reduction but at some extra cost (Huang & Russell, 2011).

**GNNs** are the dominant approach to applying machine learning to irregularly structured data (Bronstein et al., 2017; Battaglia et al., 2018). There has been a proliferation of architectures inspired by spectral graph theory (Defferrard et al., 2016), convolutional (GCN) (Kipf & Welling, 2022), message passing (MPNN) (Gilmer et al., 2017) and atten-

tional (GAT) (Veličković et al., 2018) approaches. More recently a range of differential equation inspired architectures (Chamberlain et al., 2021b;a; Giovanni et al., 2023) apply analytical tools to solve known problems with GNNs including stability, over smoothing and bottleneck phenomena. This algorithmic alignment along with the powerful message passing paradigm provide new solutions to some of the most pressing problems in science, including protein folding (Jumper et al., 2021), weather prediction (Lam et al., 2023), dynamics learning (Pfaff et al., 2023) and new numerical PDE solvers (Brandstetter et al., 2022; Lienen & Günnemann, 2022; Alet et al., 2019).

**Fast ML based methods** reduce the significant computational cost of classical methods for adaptive meshing. This includes work on $h$-adaptive mesh refinement (Foucart et al., 2023; Freymuth et al., 2023), and many contributions to $r$-adaptivity based on surrogate ML solvers of classical mesh movement PDEs (Yang et al., 2023; Hu et al., 2024) and supervised learning for mesh adaptivity using Graph Neural Networks (GNNs) (Song et al., 2022). A notable recent development is the universal mesh movement network (UM2N) (Zhang et al., 2024), which achieves error reduction on par with MA, but at significant speed up, and can also be applied to multiply connected domains.

# 3. Preliminaries and background

## 3.1. Problem specification

We consider finite element solutions to nonlinear second-order PDEs on general domains $\Omega$ of dimension $d$. In abstract form, we can write these PDEs as follows:

$$\mathcal{F}(u_t, u, \nabla u, \nabla^2 u) = f \quad \text{in } \Omega, \quad \alpha u + \beta \partial_n u = g \quad \text{on } \partial\Omega,$$
(1)

where $\alpha, \beta \in \mathbb{R}$. For transient problems, $\Omega = \widetilde{\Omega} \times (0, T)$, where $\widetilde{\Omega}$ is a $(d-1)$-dimensional spatial domain and $(0, T)$ is the time-interval of interest. In this case, we employ the method of lines and combine the FEM with suitable time-stepping schemes (Hairer & Wanner, 1996). To compute finite element solutions, we introduce a mesh $\mathcal{T}$ of the spatial domain with $N_z$ nodes, which we collect in the node set $\mathcal{Z}$. The mesh $\mathcal{T}$ is used to construct trial and test functions with local support to discretize (1). For example, for a Poisson problem with homogeneous Dirichlet boundary conditions we consider the space of piecewise linear functions (vanishing on $\partial\Omega$) $S_{\mathcal{Z}}$ on $\mathcal{T}$ and solve: Find $U_{\mathcal{Z}} \in S_{\mathcal{Z}}$ such that

$$(\nabla U_{\mathcal{Z}}, \nabla v)_{L^2(\Omega)} = (f, v)_{L^2(\Omega)} \quad \forall v \in S_{\mathcal{Z}};$$
(2)

where $(\cdot, \cdot)_{L^2(\Omega)}$ denotes the inner-product in $L^2(\Omega)$.

To minimise the error $E(\mathcal{Z}, U_{\mathcal{Z}})$ between the exact solution $u$ of (1) and its finite element approximation $U_{\mathcal{Z}}$, $r$-adaptive

meshing modifies the location of the node coordinates. Often, and in this work, $r$-adaptivity is particularly concerned with the reduction of the squared $L^2$-error

$$E(\mathcal{Z}, U_{\mathcal{Z}}) := \|U_{\mathcal{Z}} - u\|_{L^2(\Omega)}^2.$$
(3)

For transient problems, we tacitly assume that (3) is evaluated at the final time $t = T$.

## 3.2. Adaptive Meshing

Relocation based $r$-adaptivity turns a mesh with a certain topology into another mesh with the same topology. For this, the mesh *points* $\mathbf{z}^{(i)}$ are moved, but their *connectivity* (and hence the associated data structures) is unaltered. Such methods typically map a fixed mesh in a *computational domain* (i.e. a representation of the mesh graph) to a *deformed mesh* in the *physical domain* where the PDE is posed. Note that, for transient problems, $\Omega$ should be replaced by $\widetilde{\Omega}$ and $d$ should be replaced by $d-1$ in the following explanation (cf. Section 3.1). We denote the mesh points in the physical domain by $\mathcal{Z} = \{\mathbf{z}^{(i)}\}_{i=1}^{N_z}$ which form a triangulation $\mathcal{T}$ of $\Omega$ with $N_{\mathcal{T}}$ mesh elements, i.e.

$$\mathcal{T} \subset \left\{ \Delta^{(j)} \subset \mathcal{Z}; |\Delta^{(j)}| = d + 1 \right\}, \quad |\mathcal{T}| = N_{\mathcal{T}}.$$

We define the following domains and coordinates: the "computational" domain $\Omega_C$ is mapped to the "physical" domain $\Omega_P \subseteq \mathbb{R}^d$, and the "computational" coordinates $\boldsymbol{\xi} \in \Omega_C$ are mapped to the "physical" coordinates $\boldsymbol{z} \in \Omega_P$. To construct an adaptive mesh we consider a differentiable, possibly time-dependent, *deformation map* $\mathbf{F} : \Omega_C \to \Omega_P$, so that $\boldsymbol{z} = \mathbf{F}(\boldsymbol{\xi}, t)$ and $\mathbf{F}(\partial\Omega_C) = \partial\Omega_P$. If $\boldsymbol{\xi}^{(i)}$ are the *fixed* mesh points in the computational domain then $\boldsymbol{z}^{(i)} = \mathbf{F}(\boldsymbol{\xi}^{(i)}, t)$. Assuming the mesh in the computational domain is regular, then determining the (properties of the) mesh in the physical domain, reduces to finding, (and analysing), $\mathbf{F}$.

*Location based methods* find $\mathbf{F}$ by solving a PDE, or a linked variational principle. *Monge-Ampére* (MA) methods assume that $\mathbf{F}$ is a Legendre transform with a 'mesh potential' $\phi(\boldsymbol{\xi}, t)$ for which $\mathbf{F} = \nabla_{\boldsymbol{\xi}} \phi$. The linearisation of $\mathbf{F}$ is given by $J = \partial \mathbf{F}/\partial \boldsymbol{\xi} \equiv H(\phi)$ where $H$ is the Hessian of $\phi$. Relocation methods usually *equidistribute* a monitor function $m(z)$ so that $\phi$ satisfies the MA equation

$$m(\mathbf{z})|H(\phi)| = m(\nabla\phi)|H(\phi)| = \theta, \quad \text{for constant} \quad \theta.$$
(4)

For example, in (Huang & Russell, 2011) $m(\mathbf{z})$ is an a-priori monitor of the interpolation error. The PDE (4) has a unique, convex, solution (Budd et al., 2013) which avoids mesh tangling. However, (4) is expensive to solve and, in its pure form, only applicable to simply connected domains. Solution procedures include relaxation methods (Budd et al., 2009), quasi-Newton methods (McRae et al., 2018), surrogates (Song et al., 2022; Zhang et al., 2024), and PINNs (Yang et al., 2023).

*Velocity based methods* find an ODE describing the mesh point evolution in *pseudo-time* $\tau$ so that

$$\partial \boldsymbol{z}^{(i)}/\partial \tau = \mathbf{v}(\boldsymbol{z}^{(i)}, t). \qquad (5)$$

The choice of velocity function $\mathbf{v}$ is critical to the success of such methods, and is often motivated by natural Lagrangian structures of the underlying PDE. These methods provide the basis of our diffusion-based deformer (diffformer) described in section 4.1 and while they often lead to mesh tangling where mesh lines cross (cf. Ch. 7 in (Huang & Russell, 2011)), our architecture is specifically designed to enforce non-tangling of the mesh (cf. Theorem 4.2).

# 4. The G-adaptivity framework

The G-adaptive mesh relocation method described below is essentially a velocity based method with learnable coefficients that are trained by calculating the rate of change of the FE solution error $E$ with respect to the mesh point location. As we explain below, the G-adaptivity framework combines feature selection, structural regularization and direct optimisation to learn optimal mesh relocation in an unsupervised manner whilst avoiding mesh tangling.

## 4.1. Graph-based adaptive mesh refinement

For simplicity of exposition we focus our discussion on the 2D case, but note that this approach generalises in a straightforward manner to 3D cases as shown in Section 5.5. A mesh $\mathcal{T}$ (i.e. a triangulation of the domain $\Omega$) with meshpoints $\mathcal{Z}$ gives rise to a natural graph, with the nodeset $\mathcal{V} = \mathcal{Z}$ and the edgeset $\mathcal{E} = \{(\mathbf{z}^i, \mathbf{z}^j) \in \mathcal{V} \times \mathcal{V}; \exists \Delta \in \mathcal{T}, \text{ s.t. } \mathbf{z}^i, \mathbf{z}^j \in \Delta\}$, i.e. two nodes share an edge if there is a triangle in the mesh $\mathcal{T}$ which has both nodes as vertices. The graph $(\mathcal{V}, \mathcal{E})$ can be enriched with node features $\{\mathbf{x}_i \in \mathbb{R}^{d_0} : i \in \mathcal{V}\}$ represented in matrix notation as $\mathbf{X} \in \mathbb{R}^{N_z \times d_0}$. For example we could associate to each mesh point $\mathbf{z}^{(i)}$ the value of the solution field $u(\mathbf{z}^{(i)})$ as a feature. Likewise, we can introduce latent features that propagate through repeated application of a map on the graph, this is used in our architecture (cf. Figure 2). Mesh connectivity is stored in the adjacency matrix $\mathbf{A}$ (where $a_{ij} = 1$ if $(i, j) \in \mathcal{E}$ and zero otherwise). A graph neural network (GNN) $\mathcal{M}_\theta : \mathbb{R}^{N_z \times d_0} \times \mathcal{E} \to \mathbb{R}^{N_z \times d_N}$ is a map from features to features constructed with layers $\mathcal{L}_{\theta_k} : \mathbb{R}^{N_z \times d_k} \to \mathbb{R}^{N_z \times d_{k+1}}$ acting node wise as

$$\mathbf{x}_i^{k+1} = \mathcal{L}_{\theta_k}(\mathbf{x}_i^k) = \phi_{\theta_k}\left(\mathbf{x}_i^k, \sum_{j \in \mathcal{N}_i} \varphi_{\theta_k}(\mathbf{x}_j^k)\right)$$

where $\varphi_{\theta_k}$ is the learnable edge-wise operation, $\phi_{\theta_k}$ is a learnable node-wise aggregation and $\mathcal{N}_i = \{j \in \mathcal{V}; (i, j) \in \mathcal{E}\}$ is the set of nodes adjacent to the meshpoint $i$.

Integral to the G-adaptivity framework is the construction of the feature matrix such that the GNN can act as a mesh deformer. Similar to (Zhang et al., 2024) we construct the feature matrix by concatenating coordinates of a regular mesh $\boldsymbol{\xi} \in \mathbb{R}^{N_z \times d}$ with a learnable feature encoding $\mathbf{h}_\theta(\mathcal{Z}_0, H)$ which, motivated by (4), is dependent on the Frobenius norm of the Hessian $H(U_{\mathcal{Z}^0}) = \|\partial_i \partial_j u\|_F : \Omega \to \mathbb{R}$ of the FEM solution $U_{\mathcal{Z}^0}$ on the undeformed mesh $\mathcal{Z}^0$. When higher order finite element functions are used in the approximation space of $U_\mathcal{Z}$ the Hessian can be obtained by simple differentiation, but even in the case of linear elements this information is recoverable using widely-used techniques such as the one described in Appendix A.2. The final input feature matrix is then $\mathbf{X} = (\mathcal{Z}^0 \| \mathbf{X}_\lambda^0) \in \mathbb{R}^{N_x \times (d+|\lambda|)}$, $\mathbf{X}_\lambda^0 = \mathbf{h}_\theta(\mathcal{Z}^0, H)$, where $\lambda$ denotes the index set for the node features, which is passed into a GNN mesh deformer that then outputs the relocated mesh points. Previous works (Song et al., 2022; Zhang et al., 2024) used a graph attention network (GAT) (Veličković et al., 2018) as the GNN mesh deformer

$$\begin{pmatrix} \mathcal{Z}^{k+1} \\ \mathbf{X}_\lambda^{k+1} \end{pmatrix} = \begin{pmatrix} \mathbf{A}_\theta(\mathbf{X}^k)\mathcal{Z}^k \\ \sigma_\lambda(\mathbf{A}_\theta(\mathbf{X}^k)\mathbf{X}_\lambda^k \mathbf{W}_\lambda) \end{pmatrix} \qquad (6)$$

where $\mathbf{X} = (\mathcal{Z}, \mathbf{X}_\lambda)$ and $\mathbf{W}_\lambda$ is a learnable linear transformation matrix. To prevent mesh crossing the non-linearity and channel mixing are excluded from the positional channels in (6). In the above $\mathbf{A}_\theta(\mathbf{X}^k)$ is row-stochastic meaning that the top row of (6) corresponds to a graph-based averaging over graph neighbours. Motivated by (Chamberlain et al., 2021b) and velocity-based methods for meshpoint relocation introduced in section 3.2, in our G-Adaptive framework this average is replaced by a diffusion based deformer (henceforth referred to as *Diffformer*)

$$\dot{\mathcal{Z}}(\tau) = (\mathbf{A}_\theta(\mathbf{X}^k) - \mathbf{I})\mathcal{Z}(\tau), \quad \mathcal{Z}(0) = \mathcal{Z}^k, \qquad (7)$$

which is solved to a finite end time $\tau = \tau_{end}$ and leads to the meshpoint update $\mathcal{Z}^{k+1} = \mathcal{Z}(\tau_{end})$, i.e. an overall deformer of the form

$$\begin{pmatrix} \mathcal{Z}^{k+1} \\ \mathbf{X}_\lambda^{k+1} \end{pmatrix} = \begin{pmatrix} \mathcal{Z}(\tau_{end}) \\ \sigma_\lambda(\mathbf{A}_\theta(\mathbf{X}^k)\mathbf{X}_\lambda^k \mathbf{W}_\lambda) \end{pmatrix}. \qquad (8)$$

As before, the learnable attention $\mathbf{A}_\theta$ is row-stochastic, meaning (7) is essentially a diffusion equation on the graph $\mathcal{V}$. Further details on our Diffformer are provided in Appendix A.1. We can stack multiple layers of (8), each time varying the number of hidden feature dimensions which are updated using the second row of (8). We denote the overall GNN by the map $\mathcal{M}_\theta$ and a schematic overview of the components of $\mathcal{M}_\theta$ is provided in Figure 2.

## 4.2. Structural regularization

The architectural changes between (6) and (8) lead to several regularity properties that we refer to as structural regularization. The Diffformer based architecture has a key advantage

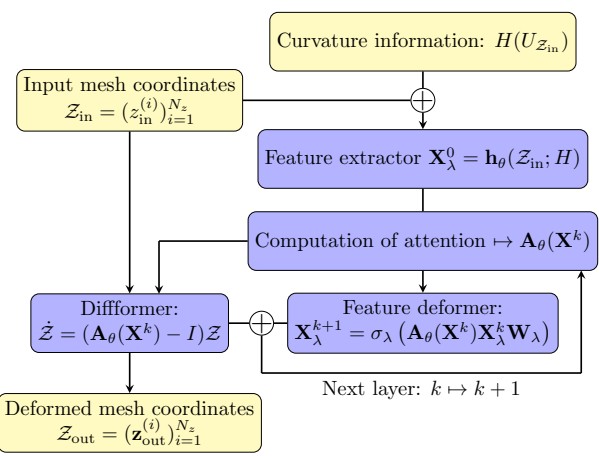

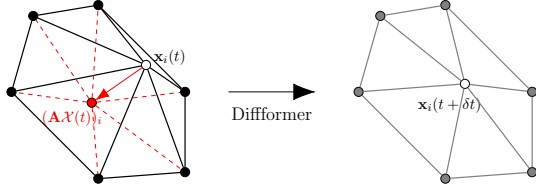

Figure 3. The action of the graph diffusion pulls nodes into the convex hull of their graph neighbours.

classical relocation methods.

### 4.3. Firedrake adjoint optimal gradient computation and direct FEM loss

Training the GNN $\mathcal{M}_\theta$ requires computing the derivative of $E(\mathcal{Z}) = E(\mathcal{Z}, U_\mathcal{Z})$ with respect to the node coordinates $\mathcal{Z}$. Since evaluating $E(\mathcal{Z}, U_\mathcal{Z})$ requires solving the PDE (1) first, a naïve application of automated differentiation would result in the solution of additional $(N_z \times d)$ PDEs $(N_z \times (d-1)$ in the transient case). To avoid the additional computational cost, we employ the well-established method of adjoints. Specifically, we employ Firedrake's automated adjoint capabilities implemented in pyadjoint (Mitusch et al., 2019). With pyadjoint, derivatives with respect to mesh coordinates can be computed in an automated fashion as shape derivatives of $E(\mathcal{Z}, U_\mathcal{Z})$ in directions discretized with vector-valued linear Lagrangian FEs (Ham et al., 2019a).

*Remark* 4.3. Often, the exact solution $u$ to (1) is not known and must itself be approximated with the FEM (e.g., by interpolating onto $U_\mathcal{Z}$ a FE solution computed on a much finer mesh). In this case, it is essential to correct the directional derivatives computed with Firedrake by adding the corrections terms stemming from evaluating the formula

$$\int_\Omega (u - U_\mathcal{Z})\nabla u \cdot V \, dx$$

along each finite element direction $V \in S_\mathcal{Z}^d$ ($V \in S_\mathcal{Z}^{(d-1)}$ in the transient case). This is necessary because the shape derivative of a FE function in a direction discretized with FEs is zero (Ham et al., 2019a).

### 4.4. Regularized gradients

In line with the concept of equidistribution discussed in Appendix E.2 we introduce a regularizing term in training which further enforces mesh regularity in an unsupervised manner and leads to improved training of the mesh deformer using only information about a predefined monitor function $m(U_\mathcal{Z})$ (we follow (Zhang et al., 2024) and use $m(U_\mathcal{Z}) = 1 + 5\frac{|\nabla U_\mathcal{Z}|}{\max_\Omega |\nabla U_\mathcal{Z}|}$). The motivation is to provide a global signal that moves mesh points into regions of the domain where the solution varies and likely requires

---

Figure 2. Schematic overview of our new graph diffusion-based architecture.

over other velocity based methods in the generation of regular meshes. A requirement of FEM meshes is that they are not 'tangled', i.e. that they form a well-posed triangulation of the domain $\Omega$ (i.e. no triangles overlap). This follows if each mesh point is in the interior of the convex hull of its neighbours on the graph and can equivalently be characterised using the Jacobian of the deformation map $\mathcal{M}$.

**Definition 4.1.** Let $\mathbf{J}^{(i)}$ be the Jacobian of the deformation map $\mathcal{M}$ at simplex $\Delta^{(i)}$. A mesh is said to be tangled if there exists a simplex where the determinant of the Jacobian $\det(\mathbf{J}^{(i)}) \leq 0$ (Huang & Russell, 2011).

Velocity based methods often lead to tangled meshes due to the local way in which the mesh point movement is defined. However, this *does not arise* in our method.

**Theorem 4.2** (Discrete-Time Non-Tangling). *If the diffusion equation* (7) *is solved with the forward Euler method, then for sufficiently small pseudo-timestep* $d\tau < 1/2$, *the discrete mesh evolution under the deformation map* $\mathcal{M}$ *preserves element orientations, ensuring that no mesh tangling occurs.*

A full proof is given in Appendix F but in essence the diffusion process ensures that meshpoints are simultaneously moved along directions that point into the convex hull of neighbouring meshpoints, thus ensuring that tangling cannot occur (cf. Figure 3).

The proof relies on the softmax of the attention mechanism normalising the adjacency to be row stochastic and for the time step of the residual connection to be controllable. This is a benefit over (6) and allows (8) to learn an anisotropic diffusion which is akin to a learnable monitor function from

more meshpoints to resolve. For this we add the following regularizing term to our loss:

$$\mathcal{L}_{\text{equi}}(\mathcal{Z}) = \sum_{\Delta^{(i)} \in \mathcal{T}} \Big| \int_{\Delta^{(j)}} m(x)dx - \overline{m} \Big|^2,$$

where $\overline{m} = |\mathcal{T}|^{-1} \sum_{\Delta^{(j)} \in \mathcal{T}} \int_{\Delta^{(j)}} m(x)dx$. Given the area of a simplex in the mesh $\alpha(\Delta^{(i)}) \in \mathcal{T}$ the terms in the above loss are approximated by $\int_{\Delta^{(j)}} m(x)dx \approx \alpha(\Delta^{(j)})c_d \sum_{\mathbf{z} \in \Delta^{(j)}} m(\mathbf{z})$, where $c_2 = 1/3, c_3 = 1/4$. This leads to the following full regularized loss which we use in the training of our Diffformer:

$$\mathcal{L}_\theta = E(\mathcal{M}_\theta) + \mathcal{L}_{\text{equi}}(\mathcal{M}_\theta) \tag{9}$$

During training the weighted graph Laplacian ($\mathbf{A}_\theta - I$ in (7)) will adaptively adjust to minimize both terms, meaning the mesh evolution will not purely follow the degree weighted graph Laplacian dynamics but will now be biased towards error reduction and equidistribution.

## 5. Experimental results

We evaluate G-adaptivity on three classical meshing problems in two-dimensions: an elliptic PDE (Poisson's equation) in a variety of convex domains, a nonlinear time-evolution PDE (Burger's equation), and the time dependent Navier-Stokes equations in a multiply connected domain. In the following we present the performance improvements obtained in terms of the FEM $L^2$-error reduction (cf. (3)) and compute time, using our novel approach for adaptive meshing on each of these problems. Additional experiments and sensitivity analysis is provide in the Appendix D. Full code to build the datasets and reproduce our results can be found at https://github.com/JRowbottomGit/g-adaptivity.

### 5.1. Experimental details

**Method** Our experimental pipeline consists of three parts: (i) we build datasets containing information about the PDE, FEM solution on a regular (i.e. not relocated) grid and the corresponding approximation of the Hessian of the solution; (ii) then we train either our model or the baseline to predict a relocated mesh on which we perform another FE solve to obtain the improved solution approximation; (iii) finally, we compare this FEM solution to a reference solution (calculated on a fine reference mesh) and determine the change in $L^2$-error over the original undeformed mesh, i.e. in the above notation we look at relative error reduction of $E(U_{\mathcal{M}_\theta})$ over $E(U_{\mathcal{Z}_0})$. These steps are repeated for the three PDE datasets as described below. Note that the Firedrake adjoint solve is only required during training of our G-Adaptive network and not during inference. The times reported in the below numerical examples thus contain a true reflection of the fast online mesh adaption times.

**Datasets** For each experiment described below we build a randomised dataset with held-out test data. This means in each case we specify a set of solution values through varying source terms or boundary conditions (adjusted to the PDE at hand). We then generate training and test sets of coarse, undeformed, meshes $\mathcal{T}$ of varying resolution with associated FEM solution values and, for each mesh, we also compute a reference solution (on a finer reference mesh) which serves as comparison for the error computation. For most test cases the solution values are generated by randomly sampling Gaussians in the domain $\Omega$. For the example of the Navier–Stokes flow around a cylinder we used snapshots of a time-series simulation of vortex shedding. Full details on the specific configurations for each experiment are provided in Appendix C.

**Baselines** We compare our algorithm against two adaptive mesh algorithms: classical MA as described and implemented by (Wallwork et al., 2024) and the ML based surrogate GNN method UM2N (Zhang et al., 2024). These two state-of-the-art approaches serve as a baseline for the FEM error reduction and deformation time. As a third baseline we train UM2N on our regularized PDE loss (9), which we denote by UM2N-G in the tables below, in order to highlight the performance improvements gained from both our new architecture (Figure 2) and our new training (9).

**Experiment details** Here we refer to our framework G-adaptivity and our model Diffformer synonymously, which we train using the regularized PDE-loss (9) (which is the regularized $L^2$ FEM approximation error + equidistribution regularizer). Calculation of the $L^2$-error is obtained by calculating an FE solution on the moved mesh and comparing this to the projection of the reference solution of a fine regular reference mesh onto sufficiently higher order elements. We train our new model (G-Adapt) and UM2N-G for 300 epochs using an Adam optimiser and learning rate of 0.001. Our model has 4 diffformer blocks as described in Appendix C. Each blocked is rolled out using explicit Euler integration for 32 steps with a step size of 0.1. For the baseline UM2N we trained using 1000 epochs in order to achieve good performance but we believe further tuning of the training may be required in order to achieve a similar performance to the one reported in (Zhang et al., 2024). UM2N remains an important baseline and we expect that with appropriate training the method would be able to achieve a similar error reduction (ER) as the Monge-Ampère (MA) solver, but we would like to highlight that even in the best reported results of the original paper, UM2N never achieved a larger error reduction than MA.

**Evaluation** We report three metrics to evaluate the performance of the mesh relocation methods at hand: (i) the relative $L^2$ error reduction (ER) of the FE solution on the

relocated mesh versus the FE solution on the initial coarse mesh (larger error reduction means improved performance); (ii) the time taken to relocate the mesh (shorter times means faster relocation); (iii) the aspect ratio of the deformed mesh as a measure of mesh quality as described in Appendix F.4 (a single digit aspect ratio is generally acceptable and a smaller aspect ratio indicates a more regular mesh). Each experiment is performed in full five times (training and evaluation) with different random seeds to provide the error bars.

## 5.2. Benchmarking on Poisson's equation

Our first benchmark is on the classical Poisson problem $-\nabla^2 u = f(\mathbf{z})$ with Dirichlet boundary conditions. Full details of the FEM formulation are given in Appendix C.1. We benchmark the results against two datasets on a square (cf. Figure 1) and polygonal domain respectively (cf. Figure 4). Further evaluations on five additional (non-convex) geometries are provided in Appendix D.4. For both aforementioned examples we sample source terms and boundary conditions corresponding to underlying Gaussian fields. On the square domain $\Omega = [0,1]^2$ we initialise the mesh-deformation with a regular grid and to showcase G-adaptivty's ability to work on irregular domains with unstructured meshes we apply a similar methodology to the convex polygonal dataset (a sample is shown in Figure 4). The results for both datasets are presented in Table 9. The central observation in these results is that our methodology provides the very first ML approach to mesh relocation which is able to outperform MA in terms of error reduction, while retaining the fast mesh relocation times given by the state-of-the-art GNN - UM2N (Zhang et al., 2024).

Table 1. Benchmarking results on Poisson Square and Poisson Convex Polygon datasets.

| POISSON SQUARE | | | |
|---|---|---|---|
| MODEL | ERROR RED. (%) | TIME (MS) | ASPECT |
| DIRECTOPT[†] | 27.40 ± 0.00 | 126028 | 33.99 ± 0.00 |
| MA | 12.69 ± 0.00 | 3780 | 2.11 ± 0.00 |
| UM2N | 6.83 ± 1.10 | 70 | 1.99 ± 0.03 |
| UM2N-G | 16.40 ±2.65 | 30 | 2.61±0.17 |
| G-ADAPT | **21.01 ± 0.33** | 88 | 2.92 ± 0.03 |
| POISSON CONVEX POLYGON | | | |
| DIRECTOPT[†] | 20.51 ± 0.00 | 56280 | 2.07 ± 0.00 |
| MA | 10.97 ± 0.00 | 4446 | 1.95 ± 0.00 |
| UM2N | 3.12 ± 0.38 | 36 | 1.66±0.03 |
| UM2N-G | 15.00 ± 0.13 | 16 | 1.88 ± 0.03 |
| G-ADAPT | **16.84 ± 0.10** | 55 | 1.86 ± 0.02 |

† The direct optimization method is included here purely for exposition, showing that MA-meshes are not necessarily optimal. DirectOpt computes the optimal mesh for a given PDE with known solution but is extremely slow and relies on

data which is not available during inference, thus it does not constitute a practical adaptive meshing strategy. In contrast, once trained, our G-Adaptive approach yields fast online mesh movement without needing reference solution values.

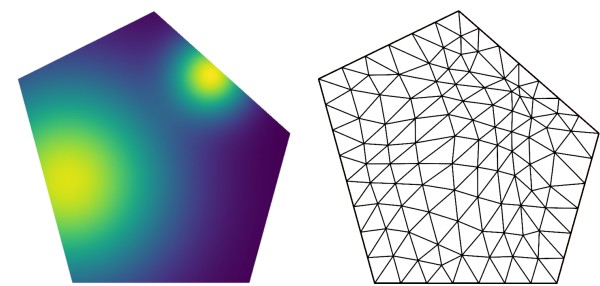

Figure 4. The solution fields and corresponding G-Adaptive mesh with 23.10 % error reduction in 59ms on a polygonal domain.

## 5.3. Time-dependent Burgers' equation

In our second example we highlight that our approach can equally well be applied to time-dependent problems, in particular the viscous Burgers' equation:

$$\frac{\partial \mathbf{u}}{\partial t} + (\mathbf{u} \cdot \nabla)\mathbf{u} - \nu\nabla^2\mathbf{u} = 0.$$

Further details on the specific FEM implementation (and implicit time-stepper) used are given in Appendix C.2. We randomly sample Gaussians on the square domain $\Omega = [0,1]^2$ as initial conditions for the evolution in Burgers' equation and perform the following experiments.

**Burgers' square rollout:** We train the models on a set of Gaussian initial conditions for a timestep $\delta t = 0.02$ with 2 steps and evaluate by following 10 trajectories of randomly sampled Gaussians in the Burgers equation for 20 timesteps, remeshing after every 2 steps (cf. Figure 5 and Appendix D.3). The results in the top part of Table 2 show the average error reduction over achieved over every block of two timesteps. While the MA performs well on this task, we note that the UM2N and UM2N-G baselines appear to lead to a *negative* error reduction (i.e. an increase), which is likely due to the fact that the Burgers' equation changes the solution shape and thus trajectories will lead to out-of-distribution cases for methods that are trained only on initial conditions. Due to the structural regularity of our new approach (cf. Section 4.2) our approach is able to deal with out-of-distribution data very well, and most importantly is able to outperform MA in terms of error reduction while retaining a fast mesh relocation time.

**Burgers' square 10 steps:** The interpolation error in remeshing is significant and provides a central limitation to current mesh relocation techniques (cf. (Budd et al.,

*Table 2.* Benchmarking results on Burgers' Square datasets.

| Model | Error Red. (%) | Time (ms) | Aspect |
|---|---|---|---|
| **Burgers' Square Rollout** | | | |
| MA | $25.78 \pm 0.00$ | 18884 | $1.99 \pm 0.00$ |
| UM2N | $-11.24 \pm 2.52$ | 314 | $2.52 \pm 0.39$ |
| UM2N-G | $-2.33 \pm 3.73$ | 316 | $2.83 \pm 0.11$ |
| G-Adapt | $\mathbf{27.17 \pm 0.34}$ | 717 | $2.85 \pm 0.12$ |
| **Burgers' Square 10 Steps** | | | |
| MA | $12.28 \pm 0.00$ | 11566 | $1.99 \pm 0.00$ |
| UM2N | $3.52 \pm 0.60$ | 30 | $2.33 \pm 0.01$ |
| UM2N-G | $16.38 \pm 2.85$ | 41 | $1.83 \pm 0.10$ |
| G-Adapt | $\mathbf{21.66 \pm 3.13}$ | 93 | $2.82 \pm 0.06$ |

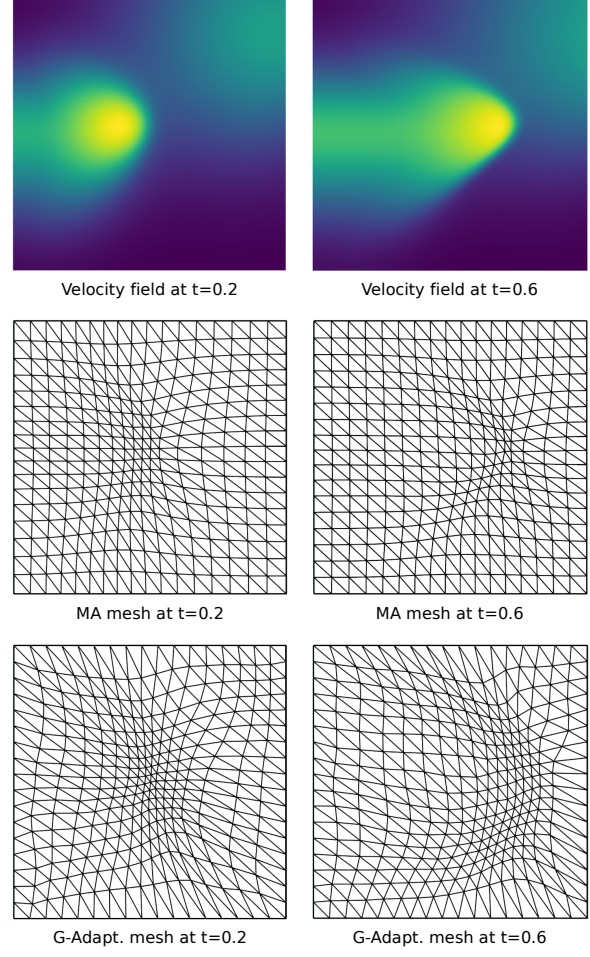

*Figure 5.* Snapshots of the velocity field (x-component) together with the corresponding deformed meshes provided by Monge–Ampère (MA) with 46.52% average error reduction over the full solution path compared to the deformed meshes provided by our approach (G-Adapt.) with 49.15% error reduction.

2009)). It is thus desirable to relocate meshes only after several timesteps. It turns out that our approach lends itself to targeted training not just of a GNN that reduces the FEM error in a stationary sense, but a GNN that seeks to find an optimal mesh *given a specified* remeshing frequency. The classical method MA has no means of inferring this information or adjusting the meshes accordingly. On this example we trained the GNN on a collection of random Gaussian initial conditions with the loss attained by solving the corresponding FEM problem for 10 timesteps of size $\delta t = 0.02$. The results in Table 2 highlight that in this way we can achieve even more significant ER over MA thus leading to efficient meshes that require less frequent changes in time-evolving systems.

### 5.4. Navier–Stokes equation and flow past a cylinder

Our final example is the canonical flow past cylinder problem we simulate data using an FE solution for 400 time steps of size $\delta t = 0.01$ of the time series evolution expressed in Gaussian basis function expansions (cf. Figure 6 and Appendix A.3). The training and test data are 25 and 50 respectively random snapshots from the range $t \in [1, 4]$ with remeshing after every 5 timesteps. Full details of the PDE and FEM formulation are provided in Appendix C.3. Again we observe good error reduction and fast mesh relocation times in our new methodology.

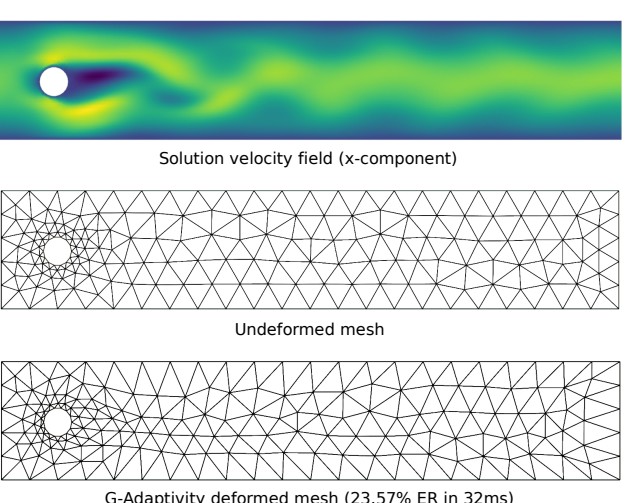

*Figure 6.* The G-Adaptivity-deformed mesh on the Navier–Stokes equation (23.57% error reduction in 32ms). The adapted mesh correctly recognises areas of large solution curvature and resolves them more finely (on the upstream side of the cylinder resolving the stagnation point singularity and along the path of shed vorticity).

*Table 3.* Benchmarking results on Navier–Stokes datasets.

| | NAVIER–STOKES | | |
| --- | --- | --- | --- |
| MODEL | ERROR RED. (%) | TIME (MS) | ASPECT |
| MA* | NA | - | - |
| UM2N† | $1.34 \pm 0.57$ | 44 | $1.65 \pm 0.06$ |
| UM2N-G | $25.55 \pm 0.81$ | 30 | $2.32 \pm 0.06$ |
| G-ADAPT | **26.36±1.37** | 49 | $3.51 \pm 0.81$ |

∗ Standard Monge–Ampère solvers do not converge on multiply connected domains. † Since no MA data is available we use the best UM2N model from Section 5.2.

### 5.5. 3D adaptive meshing

The G-Adaptivity framework and diffusion deformer model are also easily adapted to the 3D setting. To demonstrate this we perform an experiment on a 10x10x10 unit cube for the 3D Poisson problem with Dirichlet boundary conditions and Gaussian solutions, analogous to Section 5.2. An example of the corresponding results can be seen in Figure 7. In the interest of brevity, the full numerical results are presented in Appendix 5.6 and show that the method outperforms MA significantly (out-of-the-box UM2N does not apply in 3D) and that it leads to effective mesh point concentration in regions of interest.

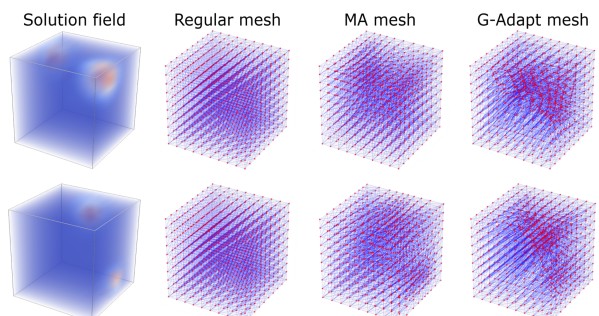

*Figure 7.* Examples of 3D solution fields and adapted meshes.

### 5.6. Scalability of the G-Adaptivity framework

The G-Adaptivity framework is able to scale to very large meshes. In particular the inductive learning property of GNNs ensures the ability of GNNs to transfer to unseen graphs in this case meaning we can perform super-resolution. In Table 9 we report experiments where the model is trained on 15x15 mesh and inference is performed on larger 60x60 (3,600 nodes) and 150x150 (22,500 nodes) meshes for the Poisson problem with 128 sampled Gaussians (see Figure 8). In order to scale the transformer encoder, which in naive form scales with $\mathcal{O}(N^2)$ edges we use a sliding window SWIN (Liu et al., 2021) style transformer to capture the monitor function embedding at the mid-length scales.

Our model consistently achieved significant mesh adaptation, accuracy improvement, and computational acceleration compared to Monge-Ampére, matching the performance observed on smaller-scale experiments.

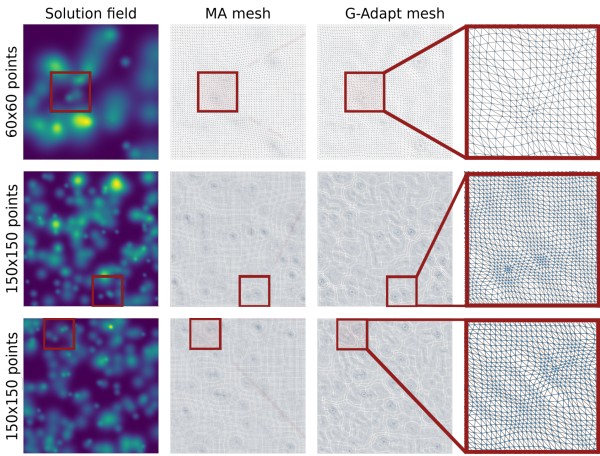

*Figure 8.* G-Adaptivity on large-scale fine meshes

## 6. Conclusions and future work

We have presented a novel, and effective, approach to the classical problem of $r$-adaptive meshing for FEM solutions of PDEs. In particular, we demonstrate, that GNNs together with a differentiable FEM solver (Firedrake), and a loss function given by the regularized solution error, can be effectively used to optimise the location of mesh points to minimise the FEM error. Hence we can take an entirely different route from prior work (both classical and ML approaches) which determine good choices of mesh points by analysis-inspired heuristics using a location based approach. We demonstrate the advantages of our method on challenging test problems in two dimensions, including in a multiply connected domain, and find that, on those examples, we are able to outperform both classical and ML methods in terms of error reduction while retaining similar computational cost to prior ML work. We note that the direct FEM error optimisation approach extends naturally to more complex domains, and PDEs, where classical methods may struggle providing a basis for future extensions of this work.

Finally, we note that any machine learning-based approach is inherently statistical in nature, meaning that GNN-based meshing tools are likely to perform worse on out-of-distribution data. We observed this in our experiments with both pre-trained UM2N models and our own G-Adaptive approach when applied to PDEs whose solutions exhibited markedly different scales and features from those seen during training. Enhancing the scale-generalisation capabilities of ML-based adaptive meshing therefore remains an important open problem for future investigation.

## Acknowledgements

The authors would like to thank Patrick Farrell and David Ham for helpful advice related to capabilities of Firedrake, as well as Joseph G. Wallwork and Mingrui Zhang for sharing code and advice on the use of the UM2N baseline model and the Python Movement package. JR, GM, TD, CBS & CB gratefully acknowledge support from the EPSRC programme grant in 'The Mathematics of Deep Learning', under the project EP/V026259/1. GM gratefully acknowledges funding from the Mathematical Institute, University of Oxford. KS gratefully acknowledges funding from the European Research Council (ERC) under the European Union's Horizon 2020 research and innovation programme (grant agreement No. 850941).

## Impact Statement

This paper presents work that aims to accelerate and improve the performance of mesh relocation methods for FEM using Machine Learning. FEMs are omnipresent in scientific computing and the generation and adaption of an effective mesh is paramount to any large-scale application of FEMs. The potential applications and benefits of advancements in this field are thus significant as complex nonlinear PDEs appear everywhere in nature: from weather and climate forecasting over oceanography up to general relativity.

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

# A. Implementation details

### A.1. Diffusion deformer (diffformer) details

We apply the diffformer in learned blocks

$$\mathcal{D}_\theta^{(b)}(\mathbf{X}) = \left( \prod_{n=0}^{T_b} (I + dt(\mathbf{A}_\theta^{(b)}(\mathbf{X}^{(b)}) - I)) \right) \mathbf{X}^{(b)}. \tag{10}$$

where $n = 0, \ldots, T_i/dt$ denotes the discrete time step index, $N_b$ is the number of blocks in the deformer, such that $\mathbf{A}_\theta^{(b)}(\mathbf{X})$ is the attentional adjacency matrix at block $b$, dynamically learned as:

$$a_{ij}^{(b)} = \frac{\exp(\phi_\theta^{(b)}(\mathbf{X}_i, \mathbf{X}_j))}{\sum_{k \in \mathcal{N}(i)} \exp(\phi_\theta^{(b)}(\mathbf{X}_i, \mathbf{X}_k))}.$$

Then the full G-adaptivity diffusion based deformation Map is given by

$$\mathcal{M}_\theta(\mathbf{X}_0, \mathbf{A}) = \left( \prod_{b=0}^{N_b} \mathcal{D}_\theta^{(b)} \right) \mathbf{X}^{(n)}, \tag{11}$$

The process consists of: 1. Initializing the feature matrix $\mathbf{X}^{(0)}$ as the feature positions. 2. Looping over $N_b$ deformer blocks, updating positions iteratively. 3. Applying $T_i/dt$ steps of discrete evolution to refine the mesh over time.

The input feature matrix is $\mathbf{X}_0 = (\boldsymbol{\xi} \,\|\, \mathbf{h}) \in \mathbb{R}^{N_x \times d + |\lambda|}$ utilises the graph transformer encoder of (Zhang et al., 2024) with the exact same hyperparameters. Similarly each attentional matrix $\mathbb{A}_\theta^{(b)}$ is adapted from the same. We use $N_b = 4$ blocks and rollout using explicit Euler time integration for 32 timesteps with a step size of 0.1.

### A.2. Hessian recovery

To identify parts of the domain $\Omega$ where the solution varies rapidly in space, we use an estimator for the local Hessian $H(x, y)$ which is inspired by the approach in (Picasso et al., 2011). For a piecewise linear function $u \in V(\Omega)$ an approximation of the components of $H$ is obtained by solving the weak problem

$$- \int_\Omega \partial_i u \partial_j v \, dx = \int_\Omega H_{ij} v \, dx \quad \text{for all } v \in V(\Omega), v|_{\partial\Omega} \tag{12}$$

for $H_{ij}$ subject to the strong Dirichlet boundary condition $H_{ij}|_{\partial\Omega} = 0$. While there might be other Hessian recovery techniques (see e.g. (Vallet et al., 2007)), we observe empirically that our approach leads to good results if the Frobenius norm $||H||_F = \sqrt{\sum_{i,j} H_{ij}^2}$ is fed as an input to the GNN.

### A.3. Gaussian basis function expansion for time-dynamic training

For technical reasons, in the Navier Stokes dataset it was necessary to provide the initial conditions used for training in analytical form as an UFL (Alnæs et al., 2014) expression that can be fed to Firedrake. To achieve this, snapshots of the pressure and velocity fields are taken at specified times during the numerical solution of the time-dependent Navier Stokes equations. The fields $w(x, y)$ are approximated as a sum of Gaussian basis functions in the form

$$w_{\text{GBF}}(x, y) = \sum_{ij} a_{ij} \phi(x - x_i, y - y_j) \quad \text{with } \phi(x, y) = \exp\left[ -\frac{1}{2} \left( \frac{x^2}{h_x^2} + \frac{y^2}{h_y^2} \right) \right] \tag{13}$$

where the $n_x \times n_y = 8 \times 8$ nodal points $(x_i, y_j)$ are arranged in a regular Cartesian grid over the domain with grid spacings $h_x$ and $h_y$. The expansion coefficients are chosen such that $w_{\text{GBF}}(x_i, y_j) = w(x_i, y_j)$. The sum on the right hand side of (13) can be implemented as an UFL expression.

## B. Notes on the use of Firedrake in G-Adaptivity

Firedrake (Ham et al., 2023) is a Python framework for the automatic solution of finite element problems. The central design idea based on composable abstractions, which allow the expression of the partial differential equation in weak form at a high level in Unified Form Language (UFL) (Alnæs et al., 2014). This abstraction is gradually lowered to generate C-kernels for matrix-assembly that can be executed in grid traversal with PyOP2 (Rathgeber et al., 2012). PETSc (Balay et al., 2019) provides a wide range of linear- and non-linear solvers for the resulting linear algebra problem. Firedrake supports a broad collection of finite element discretisations and dolfin-adjoint (Mitusch et al., 2019) allows the automatic construction of the adjoint problem for a given forward equation. The recently added interface to PyTorch (Bouziani & Ham, 2023) is crucial for the work in this paper.

### B.1. Additional details on implementation

Training the GNN requires computing the derivative of the loss function $E(Z, U_Z)$ with respect to node coordinates $Z$. Since $E(Z, U_Z)$ is a PDE-constrained functional, it is necessary to use adjoint models to compute these derivatives efficiently. The derivative and adjoint formulas depend on the loss function and its PDE constraints and automating their derivation is crucial to develop a general $r$-adaptivity methodology that can be trained seamlessly on different test cases. Firedrake is the perfect tool for this because it can derive adjoint models (Farrell et al., 2013; Mitusch et al., 2019) and automatically compute derivatives of $E(Z, U_Z)$ with respect to node coordinates (Ham et al., 2019b). Deriving these formulas by hand is nontrivial, tedious, and error prone. Firedrake is fully integrated with PyTorch (Bouziani et al., 2024), and this is key to formulate hybrid FEM-torch architectures required to train the GNN. Implementing our approach in Firedrake required minimal adaptations: the GNN model must conform to the Firedrake external operator API, and a term must be added to the derivatives with respect to node coordinates when $E(Z, U_Z)$ comprises a finite element solution computed on a finer mesh.

As an example consider the shape derivative $dJ(Z, U_Z)[T]$ of the functional $J(Z, U_Z) = ||U_Z||^2_{L_2(\Omega)}$, which is a simplified version of $E(Z, U_Z)$ in (3). The constraint on $U_Z$ is given by the simplest testcase: the weak Poisson equation in Appendix C.1 with $f = 4$. With Firedrake and PyAdjoint, we can compute $dJ(Z, U_Z)[T]$ as follows:

```
mesh = UnitSquareMesh(3, 3)
continue_annotation()
Q = mesh.coordinates.function_space()
T = Function(mesh.coordinates.function_space())
mesh.coordinates.assign(mesh.coordinates + T)
V = FunctionSpace(mesh, \CG", 1)
u = Function(V)
v = TestFunction(V)
solve((dot(grad(u),grad(v))-4*v)*dx==0, u, bcs=DirichletBC(V, 0, \on_boundary"))
J = assemble(u**2*dx)
Jred = ReducedFunctional(J, Control(T))
Jred.derivative()
```

Crucially, this only requires the implementation of the forward constraint equation in Appendix C.1. On the other hand, a tedious manual derivation of $dJ(Z, U_Z)[T]$ leads to $dJ(Z, U_Z)[T] = \int_\Omega (U_Z^2 + \nabla U_Z \cdot \nabla p - 4p)\nabla \cdot T - \nabla U_Z(DT + DT^\top)\nabla p \, dx$ with $p$ being the (weak) solution of the adjoint equation $\Delta p = 2U_Z$. These formulae are problem dependent and will be significantly more complicated for other PDE constraints. For test cases such as the Navier Stokes equations in Appendix C.3 this approach quickly becomes intractable, as highlighted in (Ham et al., 2019b, p. 1818).

In contrast, adapting the code above to the problems described in Appendix C.2 & C.3 requires only minor changes.

## C. Mathematical description of the numerical experiments

### C.1. Poisson's equation

Poisson's equation $-\nabla^2 u = f(\mathbf{z})$ is solved using the Finite Element method in the two-dimensional convex domain $\mathbf{z} \in \Omega \subset \mathbb{R}^2$. We use the weak formulation (2) and seek piecewise linear functions $u \in S_{\mathcal{Z}}$ with Dirichlet Boundary

conditions $u|_{\partial\Omega} = 0$ such that

$$\int_\Omega \nabla u \cdot \nabla v \, dx = \int_\Omega fv \, dx \qquad \text{for all test functions } v \in S_\mathcal{Z}, v|_{\partial\Omega} = 0. \tag{14}$$

### C.2. Burgers' equation

The non-linear viscous Burgers' equation describes the evolution of the vector- valued velocity field $\mathbf{u}$ as

$$\frac{\partial \mathbf{u}}{\partial t} + (\mathbf{u} \cdot \nabla)\mathbf{u} - \nu\nabla^2\mathbf{u} = 0 \qquad \text{in } \widetilde{\Omega}, \tag{15}$$

where $\nu > 0$ is the kinematic viscosity and we solve consider a two-dimensional rectangular domain $\widetilde{\Omega} \subset \mathbb{R}^2$. The term $(\mathbf{u} \cdot \nabla)\mathbf{u}$ describes non-linear convection and $\nu\nabla^2\mathbf{u}$ is the viscous diffusion.

We use a piecewise linear Finite Element discretisation with $\mathbf{u} \in S_\mathcal{Z}^{d-1}$. A simple backward-Euler timestepping method with step-size $\Delta t$ is employed to compute the velocity $\mathbf{u}_{n+1} \in S_\mathcal{Z}^{d-1}$ at the next timestep from the current velocity $\mathbf{u}_n \in S_\mathcal{Z}^{d-1}$ The time-discretised weak form of (15) is given by: find $\mathbf{u}_{n+1} \in S_\mathcal{Z}^{d-1}$ such that

$$\int_\Omega \left( \frac{\mathbf{u}_{n+1} - \mathbf{u}_n}{\Delta t} \cdot \mathbf{v} + (\mathbf{u}_{n+1} \cdot \nabla\mathbf{u}_{n+1}) \cdot \mathbf{v} + \nu\nabla\mathbf{u}_{n+1} : \nabla\mathbf{v} \right) dx = 0 \tag{16}$$

for all test functions $\mathbf{v} \in S_\mathcal{Z}^{d-1}$. The final two terms in (16) are the weak form of the nonlinear advection and viscous diffusion term respectively.

### C.3. The Navier–Stokes Equations

We consider the incompressible Navier-Stokes equations in primitive form for a time-dependent velocity field $\mathbf{u}$ and pressure $p$ in the two-dimensional spatial domain $\widetilde{\Omega} = [0, 2.2] \times [0, 0.41]$:

$$\frac{\partial \mathbf{u}}{\partial t} + (\mathbf{u} \cdot \nabla)\mathbf{u} - \nu\nabla^2\mathbf{u} + \nabla p = \mathbf{f}, \quad \text{in } \widetilde{\Omega}, \tag{17}$$

$$\nabla \cdot \mathbf{u} = 0, \quad \text{in } \widetilde{\Omega}, \tag{18}$$

Here $\nu > 0$ is again the kinematic viscosity and $\mathbf{f}$ is an external force term.

The Finite Element discretisation uses Taylor-Hood elements with piecewise linear pressure and vector-valued piecewise quadratic velocity functions $(\mathbf{u}, p) \in Q_\mathcal{Z}^{d-1} \times S_\mathcal{Z}$. The time-stepping procedure, which computes the velocity $\mathbf{u}_{n+1} \in Q_\mathcal{Z}^{d-1}$ and pressure $p_{n+1} \in S_\mathcal{Z}$ at the next timestep from the current velocity $\mathbf{u}_n \in Q_\mathcal{Z}^{d-1}$, is a variant of Chorin's projection method (Chorin, 1967; 1968). It consists of three steps, each of which requires the solution of a weak problem.

**Step 1: Compute tentative velocity $\mathbf{u}^*$**    Find $\mathbf{u}^* \in Q_\mathcal{Z}^{d-1}$ such that:

$$\int_\Omega \left( \frac{\mathbf{u}^* - \mathbf{u}_n}{\Delta t} \cdot \mathbf{v} + (\mathbf{u}_n \cdot \nabla\mathbf{u}_\text{mid}) \cdot \mathbf{v} + \nu\nabla\mathbf{u}_\text{mid} : \nabla\mathbf{v} \right) dx + \int_{\partial\Omega} (p_n\mathbf{n} \cdot \mathbf{v} - \nu(\nabla\mathbf{u}_\text{mid} \cdot \mathbf{n}) \cdot \mathbf{v}) \, ds = \int_\Omega \mathbf{f} \cdot \mathbf{v} \, dx. \tag{19}$$

for all piecewise quadratic vector-valued test functions $\mathbf{v} \in Q_\mathcal{Z}^{d-1}$ where $\mathbf{u}_\text{mid} = \frac{1}{2}(\mathbf{u}_n + \mathbf{u}^*)$. Homogeneous Dirichlet boundary conditions are applied at the top $(y = 0.41)$ and bottom $(y = 0)$ of the domain. The velocity field is prescribed on the inflow boundary at the left side of the domain as $\mathbf{u}(x = 0, y) = \left( 4.0 \cdot 1.5 \cdot y \cdot \frac{0.41-y}{0.41^2}, 0 \right)$. The weak problem in (19) is solved with a GMRES iteration that is preconditioned with successive overrelaxation (SOR).

**Step 2: Solve for pressure correction**    To ensure that the velocity field at the next timestep is divergence-free, find $p_{n+1} \in S_\mathcal{Z}$ which satisfies the following elliptic problem:

$$\int_\Omega \nabla p_{n+1} \cdot \nabla q \, dx = \int_\Omega \nabla p_n \cdot \nabla q \, dx - \frac{1}{\Delta t}\int_\Omega (\nabla \cdot \mathbf{u}^*)q \, dx \tag{20}$$

for all piecewise linear pressure test functions $q \in S_\mathcal{Z}$. To deal with the fact that the pressure is only determined up to an additive constant, homogeneous Dirichlet boundary conditions are applied to $p_{n+1}, q$ at the outflow boundary. The weak problem in (20) is solved with a conjugate gradient iteration preconditioned with algebraic multigrid (AMG).

**Step 3: Update velocity**    Find $\mathbf{u}_{n+1} \in Q_{\mathcal{Z}}^{d-1}$ such that:

$$\int_{\Omega} \mathbf{u}_{n+1} \cdot \mathbf{v} \, dx = \int_{\Omega} \mathbf{u}^* \cdot \mathbf{v} \, dx - \Delta t \int_{\Omega} \nabla(p_{n+1} - p_n) \cdot \mathbf{v} \, dx. \tag{21}$$

for all piecewise quadratic vector-valued test functions $\mathbf{v} \in Q_{\mathcal{Z}}^{d-1}$. The weak problem in (21) is solved with a conjugate gradient iteration preconditioned with SOR.

## D. Further details of numerical experiments

### D.1. Model and data hyperparameters

Table 4 shows for each PDE and geometry the number of train and test set samples as will as the resolution or node count for the train, test dataset and evaluation mesh.

| PDE | Poisson | | Burgers | Navier-Stokes |
|---|---|---|---|---|
| **Domain** | Square | Polygonal | Square | Cylinder |
| **Train/Test Samples** | 100/100 | 100/100 | 100/100 | 25/50 |
| **Train Resolution** | [15x15, 20x20] | 114 nodes | [15x15, 20x20] | 201 nodes |
| **Test Resolution** | [12x12,...,23x23] | 114 nodes | [12x12,...,23x23] | 201 nodes |
| **Eval Resolution** | 100x100 | 228 nodes | 100x100 | 402 nodes |

*Table 4.* Summary of PDE problem setups, including domains, sample sizes, and training/testing/evaluation resolutions.

## D.2. Additional results in the Poisson square and polygon case

In Figures 9 and 10 we present additional plots from the Poisson experiments on the square and polygonal domain detailed in the main paper exhibiting the types of meshes generated with our novel G-Adaptive methodology.

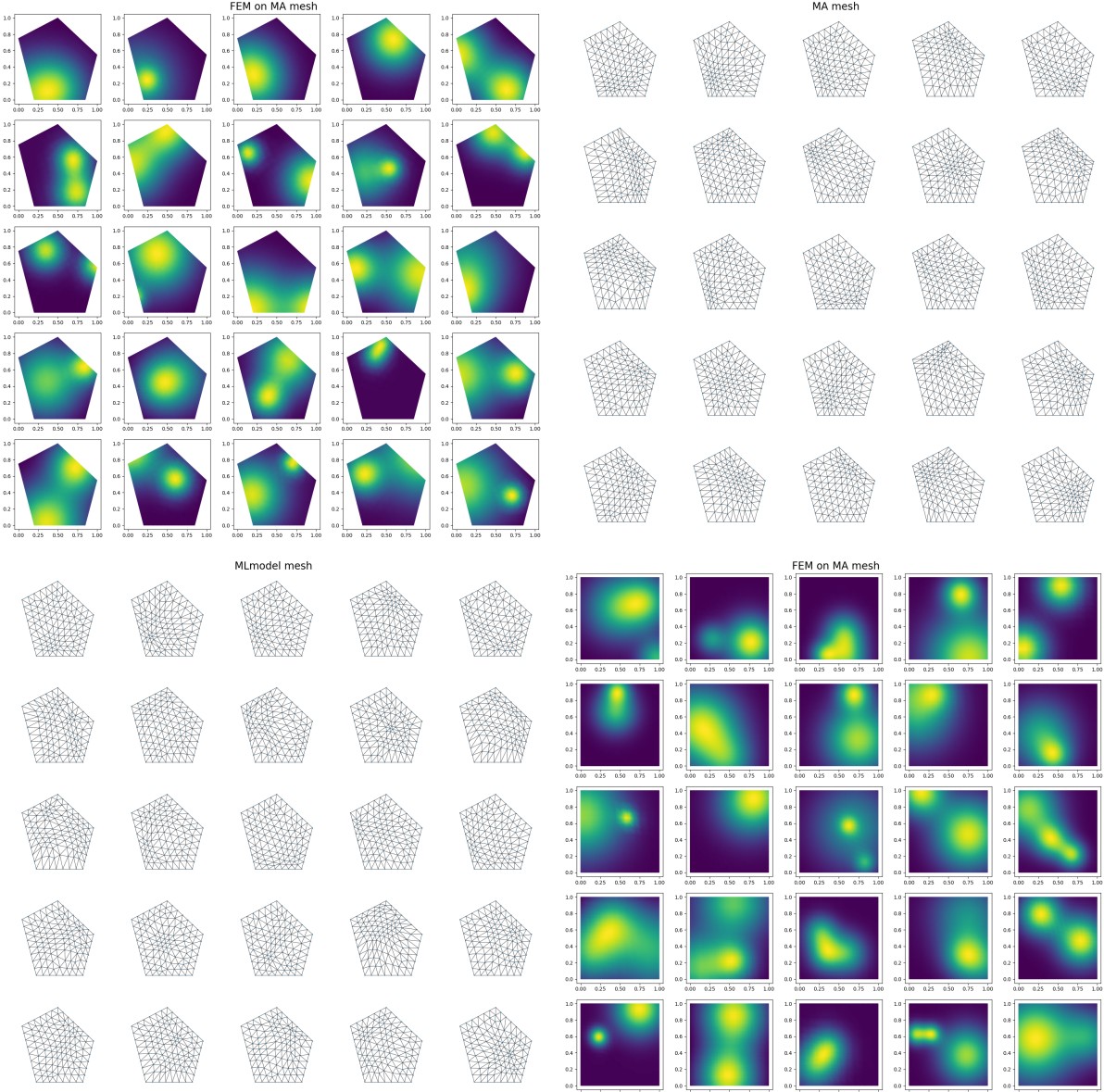

*Figure 9.* Comparison of MA and ML model-generated meshes for Poisson problems on square and polygonal domains

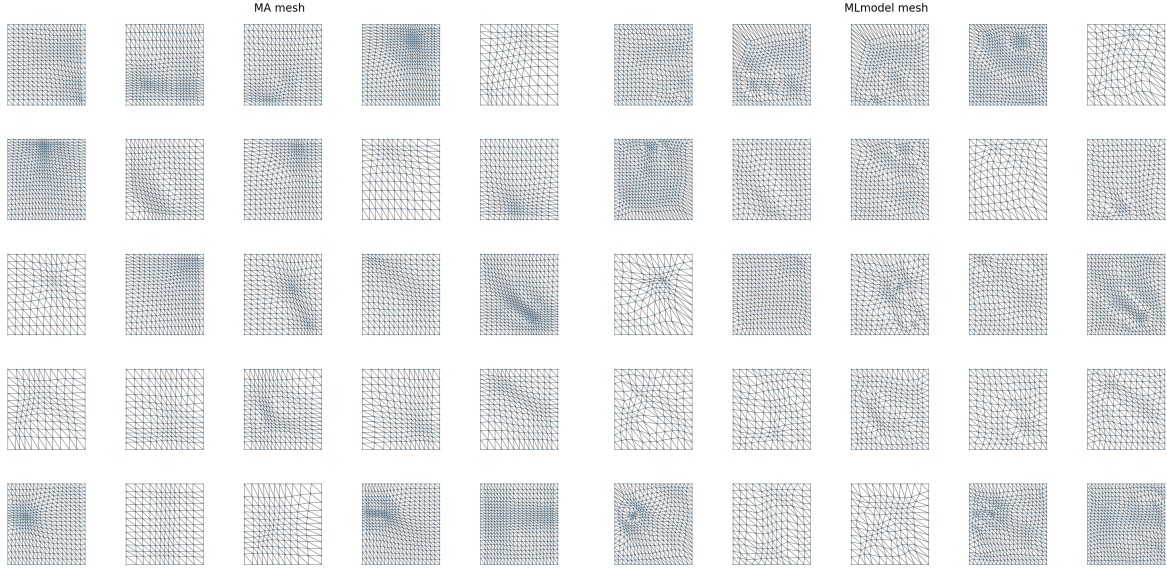

*Figure 10.* Comparison of MA and ML model-generated meshes for Poisson problems on square and polygonal domains

### D.3. Additional examples of Burger's evolution

In Figure 11 we include some additional mesh trajectories from the experiment performed in Section 5.3.

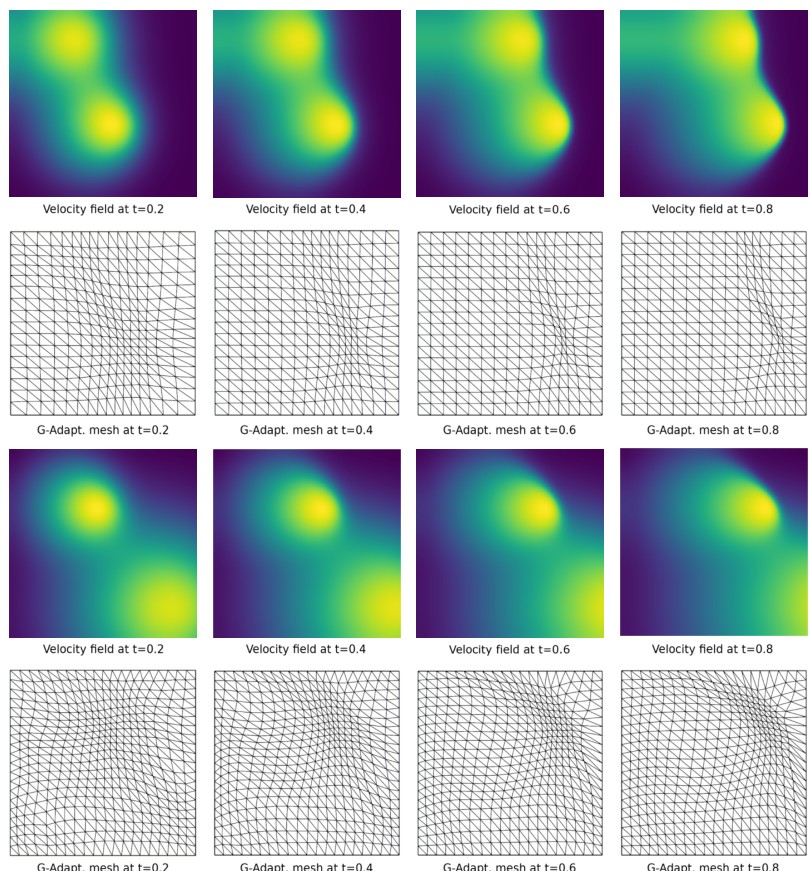

*Figure 11.* Snapshots of the velocity field (x-component) together with the corresponding deformed meshes provided by Monge–Ampère.

## D.4. Further experiments on non-convex domains

While section 5.4 already contains an example of a non-convex domain we provide further evidence that our method extends to this case using domain data from UM2N (Zhang et al., 2024). In particular we conducted experiments similar to the setup of 5.2 on five non-convex domains, cf. Figure 12. On each domain we solve Poisson's equation for randomly sampled Gaussian solutions with 100 training datapoints and 100 unseen test datapoints.

The results (error reduction scores are listed in in Table 5 and the full results can be seen in Figure 12) confirm that our method performs robustly on non-convex geometries, achieving significantly greater error reduction than baselines and generating regular non-tangled meshes on all tested domains, succeeding even when some other approaches fail. Note that the UM2N results reported below were obtained using the pretrained model from the UM2N repository, since the MA meshes obtained using (Wallwork et al., 2024) were unsuitable for direct training in these cases.

*Table 5.* Error reduction scores (%) on Poisson's equation in non-convex domain.

| DOMAIN | MA$^*$ | UM2N$^\dagger$ | UM2N-G | G-ADAPT |
|---|---|---|---|---|
| GEOMETRY 1 | $0.23 \pm 0.00$ | $-76.85 \pm 0.00$ | $1.92 \pm 0.02$ | $\mathbf{7.97 \pm 0.04}$ |
| GEOMETRY 2 | $-1.00 \pm 0.00$ | $-83.88 \pm 0.00$ | $0.69 \pm 0.08$ | $\mathbf{8.88 \pm 0.24}$ |
| GEOMETRY 3 | $-$ | $-75.82 \pm 0.00$ | $-0.96 \pm 0.04$ | $\mathbf{6.62 \pm 0.09}$ |
| H-GEOMETRY | $-108.31 \pm 0.00$ | $-73.59 \pm 0.00$ | $-0.92 \pm 0.00$ | $\mathbf{7.51 \pm 0.00}$ |
| L-GEOMETRY | $-89.40 \pm 0.00$ | $-138.43 \pm 0.00$ | $13.94 \pm 1.18$ | $\mathbf{16.25 \pm 0.25}$ |

$*$ Monge–Ampère solvers in general struggle with non-convex domains. $\dagger$ Since the MA data available is not suitable for accurate training we use the pretrained model from (Zhang et al., 2024) for our evaluation.

## D.5. Model hyper-parameter sensitivity analysis

We have performed extensive sensitivity studies and found that our approach is robust to the particular choice of hyperparameters for the diffformer blocks. We used $N_b = 4$ blocks and rollout using explicit Euler time integration for 32 timesteps with a step size of 0.1. It should be noted that the hyperparameters were identical in all experiments performed in the paper and did not require finetuning to the specific problem. Tables 6 and 7 show the sensitivity of the model to the diffusion parameters in terms of error reduction and inference time.

*Table 6.* The effect of pseudotime and diffusion timesteps on the Error reduction.

| DIFFUSION TIME AND TIME-STEP PERFORMANCE SENSITIVITY | | | | | | |
|---|---|---|---|---|---|---|
| $d\tau \setminus$ NO.-TIMESTEPS | 2 | 4 | 8 | 16 | 32 | 64 |
| 0.05 | 10.41 | 16.60 | 18.95 | 15.85 | 21.82 | 22.93 |
| 0.1 | 12.97 | 14.52 | 20.68 | 15.71 | 20.27 | 20.85 |
| 0.25 | 19.54 | 19.11 | 22.30 | 19.94 | 23.11 | 22.09 |
| 0.5 | 20.43 | 22.65 | 22.14 | 22.42 | 21.32 | 21.92 |
| 1 | 20.60 | 21.57 | 21.16 | 22.10 | 19.71 | 19.40 |

*Table 7.* The effect of pseudotime and diffusion timesteps on inference time (ms).

| DIFFUSION TIME AND TIME-STEP INFERENCE TIME | | | | | | |
|---|---|---|---|---|---|---|
| $d\tau \setminus$ NO.-TIMESTEPS | 2 | 4 | 8 | 16 | 32 | 64 |
| 0.05 | 60 | 44 | 46 | 116 | 65 | 247 |
| 0.1 | 54 | 42 | 79 | 61 | 86 | 208 |
| 0.25 | 41 | 40 | 48 | 56 | 119 | 108 |
| 0.5 | 50 | 58 | 49 | 92 | 125 | 158 |
| 1 | 52 | 59 | 45 | 69 | 100 | 108 |

We investigated the sensitivity of G-Adaptivity to the weighting of the equidistribution regularizer in Section 4.4. We found

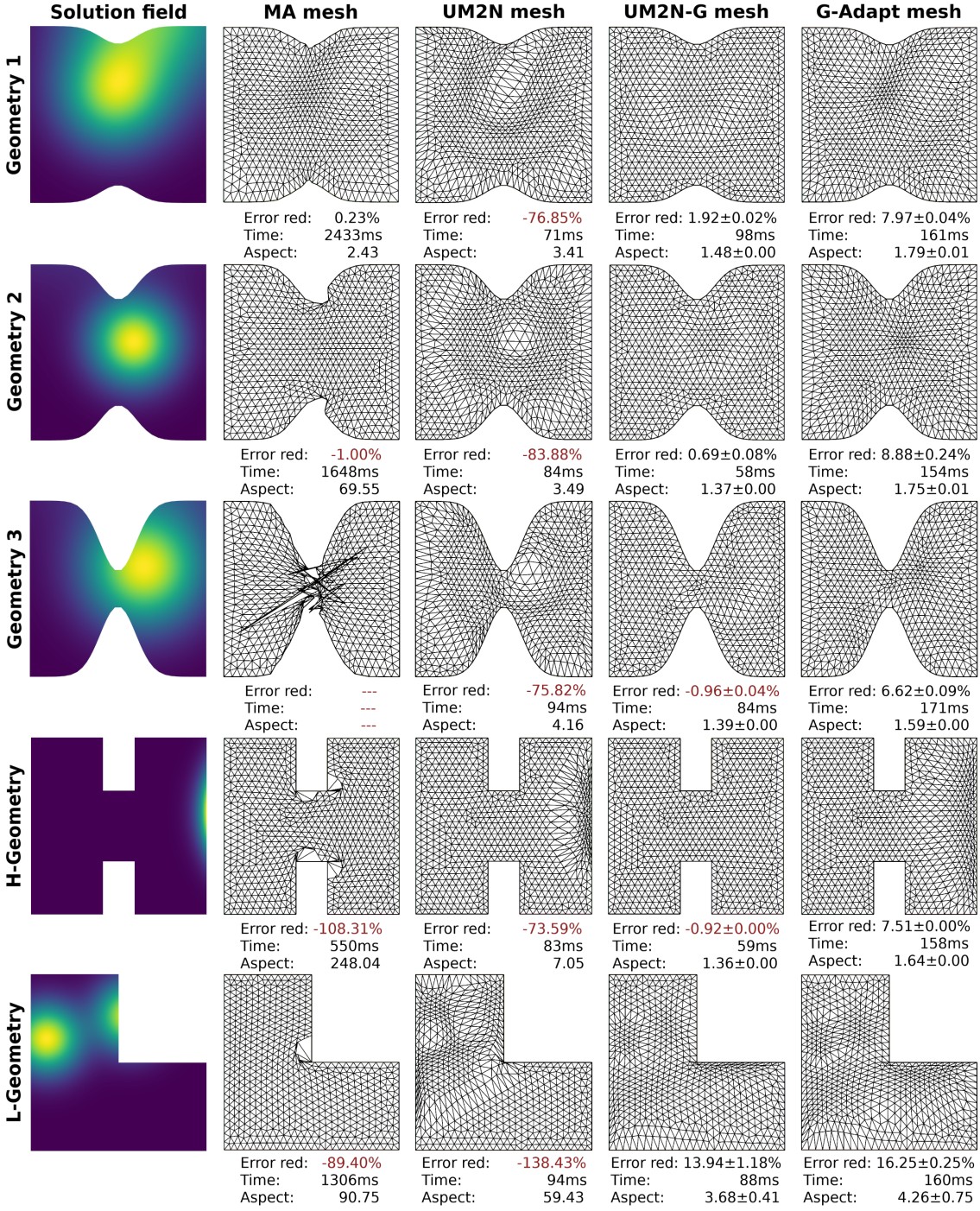

*Figure 12.* Benchmarking results for Poisson's equation on non-convex geometries.

an optimal value in the Poisson square example was a loss weighting of 1, which was used throughout all experiments.

*Table 8.* The effect of equidistribution loss regularisation on Error reduction.

| EQUIDISTRIBUTION LOSS REGULARIZER PERFORMANCE SENSITIVTY | | | | | | |
|---|---|---|---|---|---|---|
| REG. WEIGHT | 0 (NO EQUI.-DIST. LOSS) | 0.5 | 1 | 2 | 4 | 8 |
| ERROR RED. (%) | 22.42 | 22.95 | 23.99 | 23.21 | 22.14 | 20.96 |

## D.6. Scalability of the G-Adaptivity framework

In Section 5.6 we claimed the G-Adaptivity framework is able to scale to very large meshes via super-resolution. In Table 9 we report experiments where the model is trained on 15x15 mesh and inference is performed on larger 60x60 (3,600 nodes) and 150x150 (22,500 nodes) meshes for the Poisson problem with 128 sampled Gaussians (see Figure 8). We also show the results for G-Adaptivity applied to a 10x10x10 (1,000 node) cube (see Figure 7).

*Table 9.* Benchmarking results of scaling G-Adaptivity on Poisson Square and Cube datasets.

| SCALING G-ADAPTIVITY | | | |
|---|---|---|---|
| SCALE | MODEL | ERROR RED. (%) | TIME (MS) |
| 60x60 | MA | 11.94 ± 1.50 | 23084 |
| | G-ADAPT | 27.47 ± 0.89 | 452 |
| 150x150 | MA | 17.96 ± | 115395 |
| | G-ADAPT | 25.70 ± 1.51 | 2555 |
| 10x10x10 | MA | 12.71 ± 0.00 | 41049 |
| | G-ADAPT | 28.08 ± 0.36 | 494 |

# E. Further details on mesh relocation

## E.1. Advantages and limitations of $r$-adaptivity

$r$-adaptivity is a newer technology than $h$-adaptivity and as such is not yet widely adapted in industry. However, it has certain significant advantages over $h$-adaptivity. In particular it works with a constant data structure, is easy to use on parallel architectures, it gives a more regular mesh (often with guaranteed mesh regularity), it naturally inherits Lagrangian and scaling structures in a PDE (which is very useful for example in ocean modelling and studying PDEs with singularities), and can be easily linked to existing external software designed to solve a PDE on an unstructured mesh (for example a discontinuous Galerkin solver). As a result, $r$-adaptive methods have recently been very successfully used, for example, in the operational data assimilation codes of national weather forecasting offices, which when coupled to the computational dynamical core, have led to a very significant increases in computational accuracy, particularly for resolving local weather features such as fog and ice (Piccolo & Cullen, 2012). $r$-adaptivity has also found natural applications in the steel industry where the Lagrangian nature of the approach is very well suited to the evolving fine structures in the forging process (Uribe et al., 2024). Possible disadvantages of $r$-adaptivity, such as excessive mesh computation cost, and a tendency to mesh tangling, are exactly the issues we address in this paper, proposing a fast and accurate method which avoids tangling.

## E.2. The Equidistribution Principle

The equidistribution principle applied to a mesh with cells $C_i$ used for an FE calculation of a function $u(\mathbf{z})$, aims to *minimise* the total error over all the cells by equidistributing it over each cell. Typically the error over such a cell can be measured (or estimated) by the integral of an appropriate monitor function over that cell, or more simply by the expression

$$m(\mathbf{z})|C_i| \tag{22}$$

where $\mathbf{z}$ is a representative point in the cell, and (in the two dimensional case) $|C_i|$ is the cell area. The equidistribution condition on the cells $C_i$ then becomes

$$m(\mathbf{z})|C_i| = \theta, \tag{23}$$

where $\theta$ (to be determined) is a constant. The function $m$ is usually a function of $u$. An important example is given by the problem of linearly interpolating $u(\mathbf{z})$ as it follows from Céa's lemma that the resulting interpolation error is an (often tight)

upper bound for the FE solution error. In this case $m$ will be a function of the curvature of $u$ (with the exact form dependent on the norm used to measure the error) (Huang & Russell, 2011).

In the context of $r$-adaptivity each such cell $C_i$ in the *physical* domain, will be the image, under the action of the deformation map $\mathbf{F}$ of a reference cell (of fixed area) in the *computational* domain. The area $|C|_i$ of $C_i$ will then be proportional to $\det(J)$ where $J$ is the Jacobian of $\mathbf{F}$. The equidistribution condition (23) then becomes:

$$m(\mathbf{z})\det(J) = \theta. \tag{24}$$

Note that the application of the monitor function in this way is equivalent to defining a *measure* on the physical space.

In one dimension the equidistribution condition (24) uniquely defines each cell length, and thus the cell shape, and hence the whole mesh. However in two dimensions it only gives the cell area but not the shape. To find the mesh uniquely additional conditions must be imposed. Noting the correspondence between the equidistribution condition and a measure on the physical space, the deformation map can be viewed as mapping a uniform measure in the computational space to a new measure in the physical space. It is natural to seek a map which minimises the cost of doing this, as this leads to meshes in the physical domain which are close to uniform and hence have minimal skewness and which avoid tangling. This gives an obvious link between mesh generation and optimal transport. In the continuous setting such a map can be calculated by solving (either directly or by using a surrogate solver) an associated Monge-Ampére equation, leading to the MA methods described in the main body of the text. Note that with modifications this procedure can also be used to generate meshes on non-planar manifolds (McRae et al., 2018).

## F. Mesh Tangling Prevention

We provide a formal proof that a mesh evolution scheme based on the row-stochastic weighted graph Laplacian does not lead to tangling, provided a sufficiently small time step is chosen. The argument follows from the positivity of the determinant of the Jacobian of the deformation, which is preserved due to the eigen-structure of the graph Laplacian.

*Remark* F.1 (Iterative Application in GNN Blocks). The below results extend to our GNN-based mesh deformer, which applies diffusion blocks iteratively. At each iteration, the network updates the node positions while resetting the adjacency weights and initial state $X_0$. Since each block follows the same form the results can be applied recursively. This ensures that stability and mesh preservation hold across multiple diffusion steps, allowing controlled adaptation of the mesh throughout the G-adaptivity pipeline.

**Definition F.2** (Weighted Random Walk Normalized Graph Laplacian). Given a weighted graph $\mathcal{G} = (\mathcal{V}, \mathcal{E}, \mathbf{A}_\theta)$ with adjacency matrix $\mathbf{A}$ and a learnable weight matrix $\mathbf{A}_\theta$, where $(A_\theta)_{ij}$ represents the weighted edge between nodes $i$ and $j$, the weighted degree matrix is defined as $D_{ii} = \sum_j (A_\theta)_{ij}$. The weighted random walk normalized graph Laplacian is given by:

$$\Delta_\theta = I - \mathbf{D}^{-1}\mathbf{A}_\theta.$$

The operator $\Delta_\theta$ is symmetric positive semi-definite, satisfying $\Delta_\theta \succeq 0$. Its eigenvalues satisfy $0 = \lambda_0^{\Delta_\theta} \leq \ldots \leq \lambda_{n-2}^{\Delta_\theta} \leq \rho_{\Delta_\theta}$, with $\rho_{\Delta_\theta} \leq 2$. The eigenvalues represent the graph *frequencies*, and the corresponding eigenvectors are denoted by $\{\phi_\ell^{\Delta_\theta}\}_{\ell=0}^{n-1}$.

The weights $(A_\theta)_{ij}$ satisfy $a_{i,j} > 0$ if $(i,j) \in \mathcal{E}$ and $\sum_{j \in \mathcal{N}(i)} a_{ij} = 1$.

- In the degree-normalised graph (random walk) Laplacian ($A_\theta = A$), row sums are preserved due to the degree normalization, ensuring $\sum_j \tilde{A}_{ij} = 1$, where $\tilde{A} = D^{-1}A$.

- In the **softmax-weighted case**, weights are computed as

$$(A_\theta)_{ij} = \frac{\exp(f(X_i, X_j))}{\sum_{k \in \mathcal{N}_i} \exp(f(X_i, X_k))},$$

enforcing row stochasticity $\sum_j (A_\theta)_{ij} = 1$.

**Definition F.3** (Jacobian of mesh Deformation Map)**.** Given the mesh deformation model $\mathcal{M}_\theta : (\mathbf{X}, \mathbf{A}) \mapsto \mathcal{X}$, the Jacobian $J$ of the transformation is given by:

$$J = \nabla \mathcal{M}(\mathbf{X}),$$

where $\nabla \mathcal{M}(\mathbf{X})$ is the local derivative of the deformation map.

**Definition F.4.** (Mesh Tangling)**.** We say that a physical mesh is tangled if at least one simplex in the triangulation has a negative determinant in its Jacobian matrix, i.e., $\det(J_i) \leq 0$, for some $i$, where $J_i$ is the Jacobian matrix of the affine transformation mapping the reference element to the physical element in the mesh. Equivalently, the mesh is untangled if all eigenvalues of the Hessian of the transformation function, or its discrete counterpart given by the graph Laplacian, remain positive.

**Proof of Mesh Tangling Prevention**

We prove that a Laplacian GNN-based mesh adaptation scheme prevents tangling, given a sufficiently small time step. The argument follows from the positivity of the determinant of the mesh deformation Jacobian, which is preserved due to the eigen-structure of the graph Laplacian.

**F.1. Continuous-Time Evolution**

The evolution of node positions follows the Laplacian-based update:

$$\frac{d\mathbf{X}}{dt} = (\mathbf{A} - \mathbf{I})\mathbf{X} = -\Delta\mathbf{X}.$$

where $\Delta$ is the weighted random-walk graph Laplacian. As $A$ is frozen over every diffusion block, the solution of this ordinary differential equation is:

$$\mathbf{X}(t) = e^{-t\Delta}\mathbf{X}(0), \tag{25}$$

implying the determinant of the transformation Jacobian satisfies

$$J(t) = \det(e^{-t\Delta})J(0) = \left(\prod_i e^{-t\lambda_i}\right)J(0) = e^{-t\operatorname{tr}(\Delta)}J(0). \tag{26}$$

Since $\operatorname{tr}(\Delta) = N_x \times d \geq 0$, we have $J(t) > 0$ for all $t \geq 0$, ensuring that no elements invert.

**F.2. Time Step Constraints for Mesh Preservation**

The discrete update for the mesh is $\mathbf{X}^{k+1} = (I - dt\Delta)\mathbf{X}^k$, propagating the determinant as $J^{k+1} = \det(I - dt\Delta)J^k$. To prevent inversion, we require $\det(I - dt\Delta) > 0$.

**Theorem F.5** (Time Step Condition for Mesh Preservation)**.** *Given the discrete update $X^{k+1} = (I - dt\Delta)X^k$, the mesh remains untangled if $dt < \frac{1}{2}$.*

*Proof.* The determinant of the deformation Jacobian propagates as $\det J^{k+1} = \det(I - dt\Delta)\det J^k$. To ensure $\det J^{k+1} > 0$, we require $\det(I - dt\Delta) > 0$. The eigenvalues of $I - dt\Delta$ are $\mu_i = 1 - dt\lambda_i$ so the determinant condition reduces to

$$\prod_i (1 - dt\lambda_i) > 0.$$

Noting that $A$ has positive entries with row sum equal to 1, it follows by Gershgorin's theorem that the eigenvalues of $A - I$ are contained in the Gershgorin circle $|\lambda_i - 1| < 1$. Seeing as the coefficients of $A - I$ are real-valued the eigenvalues of $A - I$ are either real-valued or come in complex conjugate pairs. If $\lambda_i$ is real-valued the contribution to the above determinant is $1 - dt\lambda_i > 1 - 2dt > 0$ if $dt > 1/2$. If $\operatorname{Im}\lambda_i \neq 0$ then $\overline{\lambda_i}$ is also an eigenvalue and the contribution to the determinant is $(1 - dt\lambda_i)(1 - dt\overline{\lambda_i}) = |1 - dt\operatorname{Re}\lambda_i|^2 + |\operatorname{Im}\lambda_i|^2 > 0$. Hence we obtain $\det(I - dt\Delta) > 0$ if $dt < \frac{1}{2}$. $\square$

## F.3. Monitor-Conditioned Time Step

To refine the time step bound, consider the propagation matrix $M = I - dt\Delta$ with eigenvalues $\mu_i = 1 - dt\lambda_i$. The condition number of $M$ is $\kappa(M) = \frac{1 - dt\lambda_{\min}}{1 - dt\lambda_{\max}}$. Similarly, by Gershgorin's theorem, $\lambda_{\max} \leq 2$.

**Theorem F.6** (Monitor-Conditioned Time Step). *Given the discrete update $X^{k+1} = (I - dt\Delta)X^k$, where the monitor function redistributes the mesh to improve spectral conditioning, the time step satisfies*

$$dt \leq \min\left(\frac{1}{2}, \frac{\kappa(M)}{2}\right).$$

*Proof.* The local mesh determinant propagates as $J^{k+1} = \det(I - dt\Delta)J^k$. Stability requires $1 - dt\lambda_{\max} > 0$. Since $\lambda_{\max} \leq 2$, we obtain $dt \leq \frac{\kappa(M)}{2}$, completing the proof. $\qquad\square$

## F.4. Mesh Quality Measures

Mesh quality measures are often used as *indicators* of whether a mesh will be effective when used to solve a PDE. In particular, the mesh-quality metrics are directly related to the conditioning of FEM stiffness matrices, meaning poor mesh conditioning leads to numerical instabilities in the FEM solvers. Our method is designed to minimise the FE solution error *directly*, but the inclusion of the equidistribution regularisation ensures that G-Adaptivity leads to meshes that maintain good conditioning while reducing the FEM error.In our numerical experiments we report aspect ratio as a strong indicator of this mesh conditioning, but in the relevant literature the following metrics are commonly used to often used to assess mesh scale, skewness, and regularity.

Two paradigms exist for evaluating mesh quality:

- **Known deformation map**: Mesh quality is assessed directly using the eigenvalues $\lambda_0, \lambda_1$ of the Jacobian.

- **Local geometric properties**: Skewness can be measured as the ratio of the circumcircle to incircle radius, while regularity is inferred from element area variance.

Mesh quality can be quantified through:

- **Scale**: Element size, measured as $\lambda_0\lambda_1$, compared to a natural length scale.

- **Skewness**: The anisotropy of elements, given by $\lambda_1/\lambda_0$.

- **Regularity**: Consistency of adjacent elements, e.g., variance in element areas.

- **Consistency**: Stability of element shapes across the domain.

**Aspect Ratio** for our evaluation we use the aspect ratio of a triangular element, which is defined as the ratio of the longest edge $l_{\max}$ to the shortest altitude $h_{\min}$:

$$\text{AR} = \frac{l_{\max}}{h_{\min}}, \tag{27}$$

where $h_{\min}$ is the shortest perpendicular distance from the opposite vertex to the longest edge. A higher aspect ratio indicates more elongated elements.

