# OpenReview forum: "G-Adaptivity: optimised graph-based mesh relocation for finite element methods"
_ICML.cc/2025/Conference — ICML 2025 spotlightposter_

### Official Review · Reviewer_TRGq · 2025-03-10

**Overall Recommendation:** 3

**Summary:**

This paper proposes to apply graph neural network (GNN) architectures for mesh relocation in finite element methods (FEM). The technical contributions involve a novel training pipeline and improved network structures together with appropriate loss functions. Experiments demonstrate the effectiveness and potential of the proposed learning paradigm.

**Claims And Evidence:**

Yes.

**Essential References Not Discussed:**

No.

**Experimental Designs Or Analyses:**

Yes.

**Methods And Evaluation Criteria:**

Yes.

**Other Comments Or Suggestions:**

Overall, the technical scope of this paper is quite different from my expertise, making it difficult for me to provide detailed and in-depth evaluations of the specific technical implementations and experimental setups. However, I believe that the approach of applying GNNs for mesh relocation in this paper is rather valuable.

**Other Strengths And Weaknesses:**

From my point of view, applying GNN for mesh deformation in FEMs is an interesting and highly promising approach. My major concern is the scalability of GNN architectures when the number of nodes increases significantly. This issue may restrict the applicability of the proposed method in other more complicated computation scenarios. The authors should provide more detailed discussions.

**Questions For Authors:**

N/A

**Relation To Broader Scientific Literature:**

Well related to GNN and FEM literatures.

**Theoretical Claims:**

Yes.

---

> ### Author Rebuttal · Authors · 2025-03-31
>
> We appreciate your positive feedback and recognition of the importance and potential impact of our work. Your comments encouraged us to **test our model at significantly larger scales, including 3D simulations**. We hope the [additional experiments](https://imgur.com/a/rOdOAA0) we performed adequately address your concerns, further strengthen the paper, and provide sufficient grounds to upgrade your recommendation to a clear acceptance. We agree that scalability is crucial for the relevance of any adaptive meshing approach. It turns out that our approach scales very well to larger problems, in three ways.
>
> ### **Scalability of the GNN Model**
> **Firstly,** in forward mode, the diffusion deformer is able to scale to very large meshes by design. In particular the inductive learning property of GNNs ensures the ability of GNNs to transfer to unseen graphs in this case meaning we can perform super-resolution to scale to very large meshes. Our experiments demonstrate that our model can efficiently relocate **tens of thousands of nodes in just over two seconds on a standard laptop.**
>
> GNNs have been widely observed to scale well. For instance:
>
> - The "OGB-LSC: A Large-Scale Challenge for Machine Learning on Graphs" (Hu, 2021) demonstrates GNNs frequently being scaled to large graphs of millions/billions of nodes and edges.
> - Recent works like GraphCast (Lam, 2023), Aurora (Bodnar, 2024), MeshGraphNets (Pfaff, 2020) and Scalable Universal Physics Transformers (Alkin, 2025) all evidence the ability of transformer and GNN architectures to scale to very large scale simulations.
>
> To substantiate our claims, we conducted **new experiments on a larger 150x150 mesh (22,500 nodes)** for the Poisson problem with 128 sampled Gaussians (see [Figures C](https://imgur.com/a/rOdOAA0)). Our model **consistently achieved significant mesh adaptation, accuracy improvement, and computational acceleration compared to Monge-Ampere (MA), matching the performance observed on smaller-scale experiments**.
>
> |Scale|Model|Error Reduction (%)|Time (ms)|
> |-|-|-|-|
> |60x60|MA|11.94 ± 1.50|23,084|
> ||G-Adapt|**27.47 ± 0.89**|**452**|
> |150x150|MA|17.96 ± 1.27|115,395|
> ||G-Adapt|**25.70 ± 1.51**|**2,555**|
>
>
> One design challenge to note is that naive fully connected transformer encoder would create $22,500^2$ edges, making it computationally prohibitive. We overcame this by using a **sliding window (SWIN) style transformer** to capture the monitor function embedding at the mid-length scales. With this design choice the architecture can be scaled up much further thus providing a method that offers significant potential for real-life FEM applications. Moreover, for even larger-scale problems, **leveraging Firedrake and mesh hierarchy (multi-grid) techniques** could further enhance scalability, which we plan to explore in future work.
>
> ### **Extending to 3D Simulations**
> **Secondly,** to further assess **scalability for real-world applications**, we expanded our method to 3D domains, prompted by the suggestions by reviewers Kufx and MACS. In [Figure A](https://imgur.com/a/rOdOAA0) and the below table we demonstrate that the G-Adaptivity framework and diffusion deformer model are easily adapted to the 3D setting performing an experiment on a 10x10x10 unit cube for the 3D Poisson problem. These results confirm that our approach leads to **highly competitive error reduction in the 3D setting while maintaining computational efficiency.**
>
> |Model|Error Reduction (%)|Time (ms)|Aspect|
> |-|-|-|-|
> |MA|12.71 ± 0.00|41,049|2.97 ± 0.00|
> |G-Adapt|28.08 ± 0.36|494|6.91 ± 0.20|
>
> ### **Scalability of the FEM solver**
> **Finally,** the FEM solver which is used for the training of our GNN can also be efficiently scaled. **If a scalable solver is known for a particular PDE, then it can be easily adapted into the G-Adaptivity framework as Firedrake supports “sophisticated, programmable solvers through seamless coupling with PETSc”**. A good reference for this is (Kirby & Mitchell, SINUM, 2018). This allows us to leverage domain expertise and achieve scalability when this is needed. In our experiments, the default PETSc options of the Firedrake class `NonlinearVariationalSolver` suffices for competive FEM solution times.
>
> The same applies to solving the adjoint equations generated with Pyadjoint. Firedrake allows passing the additional key "adj_args" to the state solver parameters to specify which solver should be used for the adjoint equation thus enabling **scaling in the same manner as the forward solver**.

---

> > ### Comment · Reviewer_TRGq · 2025-04-08
> >
> > Thanks for the detailed responses and experiments. I have no further questions. I will maintain my rating as weak acceptance.

---

### Official Review · Reviewer_Kufx · 2025-03-10

**Overall Recommendation:** 3

**Summary:**

This paper presents a novel approach to mesh relocation (r-adaptivity) in finite element methods (FEMs). Traditional r-adaptive methods optimize mesh geometry by solving additional meshing PDEs, which are computationally expensive. Recent machine learning (ML) methods focus on learning surrogates for these classical techniques. In contrast, this work introduces a graph neural network (GNN)-based approach that directly minimizes the FEM solution error rather than relying on surrogate error estimates. The proposed G-Adaptivity framework employs a diffusion-based GNN mesh deformer, which optimizes mesh point locations while enforcing non-tangling constraints. The paper claims that this outperforms both classical PDE-based methods and previous ML approaches in terms of FE solution accuracy while maintaining the computational efficiency of ML-based solutions.

**Claims And Evidence:**

The claims made in the submission are generally well-supported by empirical evidence. However, the Firedrake adjoint optimal gradient computation requires clarification. The authors remark that when the exact solution is unavailable, they approximate it using FEM interpolation on a high-resolution mesh. However, running a high-resolution FEM solver is computationally expensive. Is this additional computation included in the reported time comparisons? If not, the time efficiency claims should be adjusted accordingly.

**Essential References Not Discussed:**

Good to me.

**Experimental Designs Or Analyses:**

The whole evaluation parts consists of three problems: the Poisson's equation, Burgers' equation and Navier-Stokes equation. The comparison with baselines considers error reduction, computational time, and aspect ratio, which are appropriate metrics. However, there are two notable gaps in the experimental design. first it lacks 3D evaluation: all of these problem are solved in the 2D space. Extending the method to 3D surfaces or volumetric meshes would better demonstrate its scalability and generalizability. Moreover, the paper does not analyze the effect of different loss function components (e.g., equidistribution loss) on performance. An ablation study would clarify the contributions of individual components.

**Methods And Evaluation Criteria:**

The proposed method is well-motivated for r-adaptive meshing. It replaces classical PDE-based relocation with a GNN trained via backpropagation through a differentiable FEM solver (Firedrake). The evaluation criteria include: 1. Error reduction: FEM L2-error reduction relative to the baseline mesh. The results show that the proposed method achieves the best error reduction across all baselines. Notably, it succeeds in improving error reduction in the Burgers' Square Rollout, where both UM2N baselines fail. 2. Computational time:  The time required for mesh relocation. The proposed method outperforms the classical Monge-Ampère (MA) approach but is 2-3× slower than UM2N-G. 3. Mesh quality:  Evaluated using the aspect ratio of deformed meshes. The proposed method performs well overall but is outperformed by some baselines in specific cases (e.g., Poisson Convex Polygon problem).

**Other Comments Or Suggestions:**

I wonder like to see additional visualizations comparing mesh evolution across timesteps and discuss failure cases where G-adaptivity may not perform well.

**Other Strengths And Weaknesses:**

This method proposes a novel GNN-based approach that directly minimizes the FEM error, avoiding heuristic error estimates. It also provides theoretical guarantees for mesh regularity and non-tangling. The method significantly improves computational efficiency over classical methods, supported by comprehensive experiments on various PDEs and mesh topologies.

However, the proposed method is slower than UM2N baselines, which is not discussed in the paper. Besides, there is no explicit ablation study to quantify the impact of different model components, and limited discussion on generalization to 3D meshes or more complex PDEs.

**Questions For Authors:**

1. How does G-Adaptivity generalize to 3D adaptive meshing? Have you tested it on volumetric meshes or 3D curved surface meshes on complex geometry?
2. How sensitive is the model to changes in hyperparameters (e.g., number of GNN layers, training iterations)?
3. Can the proposed approach be combined with h-adaptivity for further improvements?

**Relation To Broader Scientific Literature:**

This paper is related to classic r-adaptivity, ML-based PDE solvers, and graph-based learning approaches.  It highlights the key limitation of prior ML-based r-adaptive meshing (UM2N), which relies on learning surrogates rather than directly optimizing the FEM loss. However, additional discussion on potential limitations of ML-driven meshing (e.g., generalization to higher dimensions) would be beneficial.

**Theoretical Claims:**

The theoretical claims appear sound and well-supported. The proof of non-tangling (Theorem 4.2) is particularly valuable, as it ensures that mesh deformation does not lead to degenerate elements. No major concerns were identified.

One related question: while a sufficiently small pseudo-timestep dτ is theoretically guaranteed to prevent mesh tangling, what is the practical numerical threshold for dτ? Specifically, how is this value determined in experiments, and does it require tuning for different PDEs or mesh resolutions?

---

> ### Author Rebuttal · Authors · 2025-03-31
>
> Thank you for your careful review of our manuscript and for your positive comments and questions. Below we provide further clarifications on the **role of Firedrake, hyperparameter choices** and **generalisability of our approach**. We hope the results of the additional experiments and our responses to your questions are sufficient to encourage you to consider raising your score for acceptance.
>
> ### **Claims and Evidence**
> We should clarify that a **fine grid reference solution is only required during training** of the neural network, not during inference. The Firedrake adjoint computations are only invoked during the training phase and **not** during mesh adaptivity. Thus **our method is indeed as efficient as reported** (online mesh movement in a few dozen milliseconds).
>
> ### **Theoretical claims**
> The statement that $d\tau$ needs to be "sufficiently small" means that it **only needs to be smaller than 0.5**. This was shown in the original Appendix F.2 and we have now updated Theorem 4.2 to include this in the main body of the manuscript.
>
>
> ### **Supplementary Material**
> We have added a more detailed discussion of the role of Firedrake in the supplemenatry material of the revised manuscript, and include a shortened version of this here.
>
> Training the GNN requires computing the derivative of the loss function $E(Z,U_Z)$ with respect to node coordinates $Z$, using adjoint models for efficiency. Automating their derivation is essential for a general $r$-adaptivity methodology that works across different test cases. **Firedrake is ideal for this, as it derives adjoint models and computes these derivatives automatically** (Ham et al., Struct. Multidiscip. Optim., 2019). Obtaining corresponding formulas by hand is difficult: e.g., for $J(Z,U_Z) = ||U_Z||^2_{L_2(\Omega)}$, which is a simplified version of $E(Z,U_Z)$ in (3), the corresponding derivative takes the form $dJ(Z,U_Z)[T] = \int_\Omega (U_Z^2+\nabla U_Z\cdot \nabla p - 4p) \nabla\cdot T - \nabla U_Z (DT+DT^\top)\nabla p dx$ with $p$ being the (weak) solution of the adjoint equation $\Delta p = 2U_Z$. This automatic approach can then be coupled with PyTorch using Firedrake’s ML integration (Bouziani et al., arXiv:2409.06085) to enable GNN training.
>
>
> ### **Relation to Broader Scientific Literature**
> **Limitations of ML-based meshing:** The ML-based approach is inherently statistical, meaning that **GNN-based meshing tools are likely to perform worse on out-of-distribution test data.** We observed this in our experiments both with pre-trained UM2N models and our own G-Adaptive approach when applied to PDEs whose solutions featured vastly different scales and features than those on which the models were trained.
>
> ### **Questions for Authors**
> 1. Our method **extends naturally to 3D** adaptive meshing and we have performed an additional experiment on a cubic mesh with a selection of random Gaussians comparable to the experiment from Table 1 in our manuscript. The results are shown in [Figure A](https://imgur.com/a/rOdOAA0), and the following table (note out-of-the-box UM2N cannot be applied to 3D problems):
>
> |Model|Error Reduction (%)|Time (ms)|Aspect|
> |-|-|-|-|
> |MA|12.71 ± 0.00|41,049|2.97 ± 0.00|
> |G-Adapt|28.08 ± 0.36|494|6.91 ± 0.20|
>
>
> 2. We have performed extensive ablation studies and found that our approach is **not very sensitive to the choice of hyperparameters**, as long as they remain in a reasonable range. In particular, we emphasize that **all experiments in the paper used identical hyperparameters without fine-tuning** to specific problems or PDEs.
>
> #### Table 1: The effect of $d\tau$ and diff. timesteps on the Error reduction (%)
> |$d\tau$\No.-timesteps|2|4|8|16|32|64|
> |:-:|-|-|-|-|-|-|
> |0.05|10.41|16.60|18.95|15.85|21.82|22.93|
> |0.1|12.97|14.52|20.68|15.71|20.27|20.85|
> | 0.25 | 19.54 | 19.11 |22.30 |19.94 |23.11 |22.09 |
> | 0.5 | 20.43 | 22.65 |22.14 |22.42 |21.32 |21.92 |
> | 1 | 20.60 | 21.57 |21.16 |22.10 |19.71 |19.40 |
>
> #### Table 2: The effect of $d\tau$ and diff. timesteps on inference time (ms)
> |$d\tau$\No.-timesteps|2|4|8|16|32|64|
> |:-:|-|-|-|-|-|-|
> | 0.05 | 60 | 44 | 46 | 116 | 65 | 247 |
> | 0.1 | 54 | 42 | 79 | 61 | 86 | 208 |
> | 0.25 | 41 | 40 | 48 | 56 | 119 | 108 |
> | 0.5 | 50 | 58 | 49 | 92 | 125 | 158 |
> | 1 | 52 | 59 | 45 | 69 | 100 | 108 |
>
> #### Table 3: The effect of equi-dist loss regularisation on Error reduction (%)
> |Reg. weight|Error Reduction (%)|
> |-|-|
> | 0 (no equi-dist loss) | 22.42 |
> | 0.5 | 22.95 |
> | 1 | 23.99 |
> | 2 | 23.21 |
> | 4 | 22.14 |
> | 8 | 20.96 |
>
> 3. This is an outstanding suggestion and in fact part of ongoing work by the authors. It is natural to start by relocation to find an optimal meshpoint distribution followed by h-refinement in regions that require particularly close resolution, see (Dobrev et al., Eng. Comput., 2022) and (Piggot et al., Ocean Model., 2005). The flexibility of the current and ML approaches more generally makes them ideal candidates for such an hr-adaptive approach.

---

### Official Review · Reviewer_jV3d · 2025-03-13

**Overall Recommendation:** 3

**Summary:**

This paper focuses on using graph neural network to predict deformation of the computational domain in order to reduce error of solution obtained by finite element (FE) method. The philosophy follows [1]. The contribution of this paper is three-fold. 1. The authors proposed a new design on the model architecture based on flow/velocity-type method (eq. (7) and (8)). 2. In order to train the flow-type model, the training utilized a differentiable solver (Firedrake) and directly minimizing FE solution error. 3. A new regularization term is added to the loss function (eq. (9)).

The authors benchmarked on their own dataset. Particularly, their dataset is featured with convex domain, e.g., square and convex polygons.

[1] Zhang, M., Wang, C., Kramer, S., Wallwork, J. G., Li, S.,
Liu, J., Chen, X., and Piggott, M. D. Towards universal
mesh movement networks. 2024.

## update after rebuttal

The authors supplement experiments on non-convex domains and I changed my score to 3. However, as I commented in my review, why did not the author benchmark the dataset from [1] in the first place? I did not see any reason for not doing so. Then the author agreed the necessity of the comparison and conducted experiment in rebuttal. It seems to me that the design of experiment in the original version is lacking of thoughts to miss such basic and important benchmark. Without seeing a final version of the paper, the credibility of the supplemented experiment is weakened.

**Claims And Evidence:**

See below.

**Essential References Not Discussed:**

N.A.

**Experimental Designs Or Analyses:**

See above.

**Methods And Evaluation Criteria:**

A more systematic comparison between your work and UM2N [1] should be made:

1. Why not benchmarking the same dataset of UM2N [1]? I do not see how their dataset could not be your choice and why you have to choose your own dataset instead. Particularly, the dataset of UM2N includes non-convex domains. In contrast, your dataset contains only convex domains except a cylinder case for Navier-Stokes equations.

2. From your experiment, all table 1,2,3 shows that  the regularization term in loss function (9) contributes most to your method, since UM2N-G (UM2N + the loss function (9)) also has great improvements from vanilla UM2N. Essentially, as one of the core novelties of this paper, this regularization term is a density control that encourages equidistribution of mesh nodes. Why this regularization is so effective on your dataset is under-explored. Is it as effective on the dataset of [1]? It is actually quite doubtful because the density of mesh nodes should increase in the area where error is high. Equidistribution heuristically contradicts with this strategy.

Therefore, I strongly encourage the authors benchmark their method on the dataset of [1].

[1] Zhang, M., Wang, C., Kramer, S., Wallwork, J. G., Li, S.,
Liu, J., Chen, X., and Piggott, M. D. Towards universal
mesh movement networks. 2024.

**Other Comments Or Suggestions:**

There could be a potential to fully justify any advantage of your flow-based model over existing machine learning method, e.g., UM2N. For example, is your method more data efficient? Since your method directly minimizes FE solution error with a differentiable solver, what benefit can such ``hybrid'' strategy can provide?

**Other Strengths And Weaknesses:**

**Novelty** The methodology adopted in this work has certain novelty, i.e., flow-based model + training with differentiable solver.

However, there is an apparent **weakness** in your experiment since your dataset is biased to convex domain, and therefore it is not so convincing that your method is superior to the baseline UM2N.

**Questions For Authors:**

1. Is your method only applicable to convex domain? I see you have convex setup in your experiment. If yes, it'd be better to declare this requirement of convex domain clearly in your methodology as well.

**Relation To Broader Scientific Literature:**

N.A.

**Theoretical Claims:**

N.A.

---

> ### Author Rebuttal · Authors · 2025-03-31
>
> Thank you for your careful review of our manuscript and for your valuable comments and questions, following which we have performed several additional numerical experiments. **We appreciate your suggestion to benchmark more closely against UM2N, and we have now conducted extensive new experiments, including on non-convex domains.** We hope that we were able to address your concerns and that you can update your score accordingly.
>
>
> ### **Methods and evaluation criteria**
>
> 1. While we already benchmarked on examples from UM2N (Zhang et al., 2024), namely the flow past a cylinder, and on two datasets from M2N (Song et al., 2022), namely the Poisson and Burgers' equation on a square domain, **we have now obtained additional domain data from (Zhang et al., 2024).** This allowed us to conduct extensive additional experiments on **five non-convex domains** (four from the paper (Zhang et al., 2024) and an L-shaped domain). On each domain we solve Poisson's equation for randomly sampled Gaussian solutions with 100 training datapoints and 100 unseen test datapoints. The results (see [Figure D](https://imgur.com/a/rOdOAA0) and table below) confirm that our method performs robustly on non-convex geometries, achieving significantly greater error reduction than baselines and generating regular non-tangled meshes on all tested domains, succeeding even when some other approaches fail. Note that the UM2N results reported below were obtained using the pretrained model from the UM2N repository, and we continue to investigate and refine this baseline.
>
> #### Table 1: Error reduction (%) for various methods and non-convex domains (n.b. negative indicates error increase)
> |Domain|MA|UM2N|UM2N-G|G-Adapt|
> |-|-|-|-|-|
> |Geometry 1|0.23 ± 0.00|-76.85 ± 0.00|1.92 ± 0.02|**7.97 ± 0.04**|
> |Geometry 2|-1.00 ± 0.00|-83.88 ± 0.00|0.69 ± 0.08|**8.88 ± 0.24**|
> |Geometry 3|-| -75.82 ± 0.00| -0.96 ± 0.04|**6.62 ± 0.09**|
> |H-Geometry|-108.31 ± 0.00|-73.59 ± 0.00|-0.92 ± 0.00|**7.51 ± 0.00**|
> |L-Geometry|-89.40 ± 0.00| -138.43 ± 0.00|13.94 ± 1.18|**16.25 ± 0.25**|
>
> Mesh deformation times and aspect ratios can be found in [Figure D](https://imgur.com/a/rOdOAA0).
>
> 2. **Loss regularization:** We would like to clarify a key misunderstanding: the primary novelty in our loss is **not just the regularisation term in (9) but also the FEM error term** $E(\mathcal{M}_{\theta})$. Prior works relied on MSE loss to classically adapted meshes, whereas we directly minimise the FEM error, which is the **key driver of performance improvements**. The equidistribution loss is a secondary regulariser that **further enhances FEM error reduction** by ensuring that the monitor function (not the nodes) is equidistributed. This is in line with the reviewer's own intuition: "the density of mesh nodes should increase in the area where error is high", as the nodes will be distributed to areas where the curvature of the solution is high. Our experiments already highlight the advantages of our FEM loss component (UM2N vs UM2N-G). We further conducted an **ablation study** on the Poisson dataset showing that our choice of **weight 1 is optimal for this regularising loss term**.
>
> |Regularization weight|Error Reduction (%)|
> |-|-|
> | 0 (no equi-dist loss) | 22.42 |
> | 0.5 | 22.95 |
> | 1 | 23.99 |
> | 2 | 23.21 |
> | 4 | 22.14 |
> | 8 | 20.96 |
>
> ### **Other strengths and weaknesses**
> Through our extensive additional experiments we demonstrate that **our method is not restricted to convex domains**. Across all cases, **our approach significantly outperforms the two main baselines, UM2N (ML-based) and MA (classical method), in terms of FEM error reduction, while achieving comparable computational efficiency to UM2N.**
>
> ### **Other comments or suggestions**
> We appreciate the point raised concerning the motivation and advantages of our approach.
>
> 1. A **central novelty and advantage** of our method is that it optimises the FEM solution error directly, which is in contrast to prior work, including UM2N, which had designed **surrogates** to classical meshing approaches (such as MA). These classical methods rely on heuristics and cannot directly minimise the FEM error.
> 2. The flow-based approach, i.e. the diffusion deformer components in our GNN architecture are motivated by relaxation-based mesh movement and provide a **clear, quantifiable advantage over GAT-based deformers** used in UM2N as observed in our experiments (UM2N-G vs G-Adapt).
> 3. In contrast to prior work, our approach **does not use MA-solutions as a basis for training** and as such can be seen as more data-efficient. It nevertheless requires the repeated solution of the training problems during the training part (no additional solve is required during inference).
>
> ### **Questions for Authors**
> 1. Thank you for raising this question. Our method is indeed **fully applicable to non-convex domains** as demonstrated by our additional experiments. We have updated our manuscript to explicitly state this and to showcase the above results.

---

> > ### Comment · Reviewer_jV3d · 2025-04-02
> >
> > Thank you for additional experiment to show about application on non-convex domains, which can change my evaluation of the paper. However, this also introduce significant modification of your paper, which makes it hard to evaluate accurately without seeing a complete version. Thus, I decide to change my score to 3.

---

> > > ### Author Response · Authors · 2025-04-02
> > >
> > > We appreciate the reviewer’s prompt evaluation of our new experiments on non-convex domains. We would like to clarify that these additions extend our original results without modifying the algorithm, methodology, model, or hyperparameters. The experiments adhere precisely to the framework of our original submission and serve solely to further reinforce our claims.
> > >
> > > The only manuscript changes consist of the inclusion of additional domain experiments in the main body and discussion on ablation tables in the Appendix. As all ICML 2025 papers permit an extra page for rebuttal feedback, we hope this clarifies that our work remains complete.
> > >
> > > We greatly value the reviewer’s feedback and welcome any further thoughts, as well as a potential re-evaluation of their score in light of this clarification.

---

### Official Review · Reviewer_MACS · 2025-03-16

**Overall Recommendation:** 3

**Summary:**

In this work, a GNN-based mesh relocation method is proposed by directly minimizing the finite element solution error. A diffusion-based GNN-deformer is applied which can reduce mesh tangling. Experiments show the proposed method achieves lower solution error, on Poisson's, Burgers', and Navier-Stokes equation problems.

## update after rebuttal
The authors have carefully addressed the comments and suggestions from all reviewers and have provided abundant extra experimental results. I will remain my already positive score unchanged.

**Claims And Evidence:**

The improvement in model structure and loss design makes sense, and is validated by experiments and ablation studies.

**Essential References Not Discussed:**

To my knowledge, I don't see any essential references not discussed.

**Experimental Designs Or Analyses:**

The experimental designs and analyses are sound.

**Methods And Evaluation Criteria:**

- The results on larger-scale, 3D, and more complicated geometry problems will further validate the performance of the proposed method.
- In the experiments, aspect ratio is taken as a metric. However, this is not necessarily required for good meshes, especially for this work, the final target (lower FE error) is taken to supervise the training, so we may get anisotropic good meshes. Therefore, I am not sure if it is appropriate to use aspect ratio as an evaluation metric.

**Other Comments Or Suggestions:**

None.

**Other Strengths And Weaknesses:**

None.

**Questions For Authors:**

- In industry, normally people don't care about guaranteeing the topology of the mesh, as long as efficient and accurate solutions can be provided. Can the authors provide some reasons or scenarios where r-adaptation is mandatory or better than h-adaptation?
- Why is the DirectOpt method shown in Fig. 1 not compared in the experiments?
- I am not sure if the proposed mesh deformer should be called "diffusion-based" or "neural-ode-based"?

**Relation To Broader Scientific Literature:**

This work can be applied in downstream industrial physical simulations such as fluid, structural, heat simulations, etc.

**Theoretical Claims:**

I think Theorem 4.2 is solid. Note that some strong assumptions are required, such as sufficiently small timesteps. Hence in practice mesh tangling can still happen in extreme cases.

---

> ### Author Rebuttal · Authors · 2025-03-31
>
> Thank you for your careful review of our manuscript and for your valuable comments and questions, particularly for encouraging us to perform **experiments on more complex geometries and larger scale problems,** and to provide **clarification on our theoretical results and the use of ML-based r-adaptivity in industry**. We appreciate your positive feedback, hope the following responses answer your questions and encourage you to consider raising your score for acceptance.
>
> ### **Methods and evaluation criteria**
> 1. We have expanded our experiments to include **more complex, non-convex domains, and high resolution meshes** (see [Figures C & D](https://imgur.com/a/rOdOAA0)). Additionally, our model **generalises naturally to 3D**, and we tested it on Poisson's equation on the unit cube with Dirichlet BCs and Gaussian solutions. The results presented below and in [Figure A](https://imgur.com/a/rOdOAA0) show that the method outperforms MA significantly (out-of-the-box UM2N does not apply in 3D) and leads to **effective mesh point concentration in regions of interest**.
> |Model|Error Red. (%)|Time (ms)|Aspect|
> |-|-|-|-|
> |MA|12.71 ± 0.00|41049|2.97 ± 0.00|
> |G-Adapt|28.08 ± 0.36|494|6.91 ± 0.20|
> 2. We appreciate the reviewer’s note that aspect ratio is not essential for good meshes, which is a key aspect of our framework that aims to break the existing paradigm of classical meshing techniques: **minimal FEM-error meshes do not necessarily require a small aspect ratio**. While our primary loss term is the FEM error $E(\mathcal{Z},U_\mathcal{Z})$, **mesh quality also affects FEM-stiffness conditioning**, and high aspect ratios can introduce numerical instabilities in FEM solvers. We report the aspect ratio to show that G-Adaptive meshes achieves error reduction while maintaining reasonable conditioning. A clarification has been added in Appendix F.4.
>
> ### **Theoretical claims**
> The assumption that $d\tau$ needs to be "sufficiently small" to avoid mesh tangling **only requires $d\tau<0.5$**. This was shown in the original Appendix F.2 and we have now updated the statement of Theorem 4.2 to include this in the main body of the manuscript.
>
> ### **Questions**
> 1. r-adaptivity is a newer technology than h-adaptivity and as such is not yet widely adapted in industry. However, it has certain significant advantages over h-adaptivity. In particular it works with **a constant data structure, is easy to use on parallel architectures, it gives a more regular mesh** (often with guaranteed mesh regularity), it naturally **inherits Lagrangian and scaling structures in a PDE** (which is very useful for example in ocean modelling and studying PDEs with singularities), and can be **easily linked to existing external software** designed to solve a PDE on an unstructured mesh (for example a discontinuous Galerkin solver). As a result, **r-adaptive methods have recently been very successfully used**, for example, in the operational data assimilation codes of **national weather forecasting offices**, which when coupled to the computational dynamical core, have led to a very significant increases in computational accuracy, particularly for resolving local weather features such as fog and ice (Piccolo \& Cullen, Q. J. R. Meteorol. Soc., 2012). r-adaptivity has also found natural applications in the **steel industry** where the Lagrangian nature of the approach is very well suited to the evolving fine structures in the forging process (Uribe et al., Finite Elem. Anal. Des., 2024). **Possible disadvantages of r-adaptivity, such as excessive mesh computation cost, and a tendency to mesh tangling, are exactly the issues we address in this paper, proposing a fast and accurate method which avoids tangling.**
> 2. The **direct optimization method** is used in Fig. 1 **purely for exposition**, showing that MA-meshes are not necessarily optimal. DirectOpt computes the optimal mesh for a given PDE with known solution but is extremely slow and relies on data which is not available during inference. In contrast, once trained, **our G-Adaptive approach yields fast online mesh movement** without needing reference solution values. However, inspired by your comment we have added the DirectOpt results to Table 1:
> |Model|Error Reduction (%)|Time (ms)|Aspect|
> |-|-|-|-|
> |DirectOpt|27.40 ± 0.00|126,028| 33.99 ± 0.00|
> |MA|12.69 ± 0.00|3,780|2.11 ± 0.00|
> |UM2N| 6.83 ± 1.10|70|1.99 ± 0.03|
> |UM2N-G|16.40 ± 2.65|30|2.61 ± 0.17|
> |**G-Adapt**|21.01 ± 0.33|88|2.92 ± 0.03|
>
> 3. We agree that our architecture is fundamentally a Neural ODE on a graph. However, the specific form of this differential equation is crucial to the success of our method: The governing equation, $\dot{\mathcal{Z}}(\tau)=(\mathbf{A}_{\theta}(\mathbf{X}^k)-\mathbf{I}) \mathcal{Z}(\tau)$ resembles a discretized learnable diffusion equation, motivating our use of the term "diffusion-based GNN-deformer.". This terminology aligns with prior literature on similar architectures (Chamberlain et al., 2021a;b).

---

> > ### Comment · Reviewer_MACS · 2025-04-09
> >
> > Thanks for your hard work. All my concerns have been well addressed. Extra experiments have been performed to demonstrate the effectiveness of the proposed method in scenarios with larger scales, more complex geometries, and 3D. Overall, I will remain my already positive score unchanged.

---

### Decision · Program_Chairs · 2025-05-01

**Decision:**

Accept (spotlight poster)

**Comment:**

## Strengths
- Learning-based fast r-adaptivity to optimize meshes for more accurate FE simulations with GNN and diffusion, which is important in practical applications
- Novelty in the technique directly optimizing meshes using GNN and diffusion trained with the FE loss using a differential solver, Firedrake, in contrast to former learning-based approaches, which learned the FE surrogates to optimize meshes
- Solid theoretical claim about discrete time non-tangling, ensuring that the mesh tangling cannot happen if the timestep is sufficiently small (<0.5)
- Versatility to non-convex and 3D meshes

## Weaknesses
- As the authors note, the generalization capability of learning-based methods is inherently bounded by the training data. More insights into the severity of the OOD inferencing in practical applications and how to overcome them is appreciated.
- Videos showing the optimization process and the simulated results of this method and competing methods will help the community to appreciate this method more